# CE-Nav: Flow-Guided Reinforcement Refinement for Cross-Embodiment Local Navigation

Kai Yang* Tianlin Zhang* Zhengbo Wang* Zedong Chu†
Xiaolong Wu Yang Cai Mu Xu

AMAP, Alibaba Group
{yk496472, arthur.ztl, wangzhengbo.wzb, chuzedong.czd,
huanlu.wxl, yangcai.cy, xumu.xm}@alibaba-inc.com

## Abstract

Generalizing local navigation policies across diverse robot morphologies is a critical challenge. Progress is often hindered by the need for costly and embodiment-specific data, the tight coupling of planning and control, and the "disastrous averaging" problem where deterministic models fail to capture multi-modal decisions (e.g., turning left or right). We introduce CE-Nav, a novel two-stage (IL-then-RL) framework that systematically decouples universal geometric reasoning from embodiment-specific dynamic adaptation. First, we train an embodiment-agnostic General Expert offline using imitation learning. This expert, a conditional normalizing flow model named VelFlow, learns the full distribution of kinematically-sound actions from a large-scale dataset generated by a classical planner, completely avoiding real robot data and resolving the multi-modality issue. Second, for a new robot, we freeze the expert and use it as a guiding prior to train a lightweight, Dynamics-Aware Refiner via online reinforcement learning. This refiner rapidly learns to compensate for the target robot's specific dynamics and controller imperfections with minimal environmental interaction. Extensive experiments on quadrupeds, bipeds, and quadrotors show that CE-Nav achieves state-of-the-art performance while drastically reducing adaptation cost. Successful real-world deployments further validate our approach as an efficient and scalable solution for building generalizable navigation systems.

## 1 Introduction

The recent surge in mobile robotics has led to a wide array of platforms with diverse morphologies, creating a fundamental challenge in developing navigation policies that can be seamlessly deployed across multiple embodiments. Current learning-based strategies diverge broadly in their architectural choices. On one end, **end-to-end (E2E) policies** (Wang et al., 2025) attempt to map observations directly to low-level joint commands. This approach, while powerful, deeply entangles high-level planning with the robot's specific dynamics, making the policies brittle on new platforms. On the other end, **hierarchical methods** first plan a path as a sequence of waypoints(Cai et al., 2025; Doshi et al., 2025; Yang et al., 2023; Shah et al., 2023a; Yang et al., 2024; Chu et al., 2026; Hu et al., 2025; Xiang et al., 2025; Liu et al., 2025a; Chen et al., 2025; Xue et al., 2026). This decouples planning from control, but introduces a critical gap: the high-level planner operates on a simplified or idealized model of the controller, making it difficult to compensate for unmodeled dynamic effects or imperfect tracking performance.

Hierarchical **velocity planning** emerges as a more robust "middle ground" (Xu et al., 2025b; Liu et al., 2025b; Hirose et al., 2023; Liu et al., 2025c; Truong et al., 2021). It decouples high-level geometric reasoning from low-level motor control, yet provides a reactive command interface that can be trained to compensate for the underlying system's dynamics. However, this promising approach faces two fundamental bottlenecks. **First, the reliance on expert data with embodiment-specific bias.** Sourcing data from costly, embodiment-specific real-world trajectories or physics-based simulations introduces a strong bias that limits generalization and scalability. **Second, the deterministic**

---

*Equal contribution.
†Corresponding author.

**learning paradigm.** Framing navigation as a deterministic regression task fundamentally fails to capture its inherent multi-modality (e.g., turning left or right at a T-junction), leading to "disastrous averaging" behaviors.

To overcome these specific limitations, we introduce **CE-Nav**, a novel framework for Cross-Embodiment Local Navigation that achieves both low-cost transferability and high performance. Our approach is founded on hierarchical decoupling, separating the navigation task into a high-level velocity planning policy ($\pi_{\text{high}}$) and a low-level locomotion controller ($\pi_{\text{low}}$). The high-level policy operates in a universal action space of body velocity commands ($v_x, v_y, v_{\text{yaw}}$), a standard interface for many mobile robots, including quadrupeds and bipeds. This abstraction enables the learning of a transferable navigation core, which operates atop any embodiment-specific low-level controller. Our framework does not assume this controller is an ideal velocity tracker; rather, our Stage 2 refiner is explicitly trained to compensate for its specific dynamic characteristics and execution imperfections.

The training of $\pi_{\text{high}}$ follows a two-stage paradigm that disentangles embodiment-agnostic geometric reasoning from embodiment-specific dynamic adaptation.

1. **Stage 1 (Offline IL):** We train a general navigation expert ($\pi_{\text{expert}}$) that understands universal planning principles (e.g., obstacle avoidance) purely from a kinematic perspective, **without relying on any real robot data**. Critically, we use a conditional normalizing flow model, **VelFlow**, to learn the full distribution of expert actions, effectively **resolving the multi-modality issue**.

2. **Stage 2 (Online RL):** For a new robot, we freeze the general expert and train a lightweight, **Dynamics-aware Refiner**. Guided by the expert's proposals, this refiner quickly learns to translate the general plan into dynamically feasible and optimal commands for the specific robot through minimal interaction with the environment.

This modular, plug-and-play design allows CE-Nav to endow new robotic platforms with sophisticated navigation capabilities through a brief and stable training process. Our contributions are:

- We propose a novel IL-then-RL framework that decouples universal geometric reasoning from embodiment-specific dynamics. The framework uses a multi-modal kinematic expert, trained offline on classical planner data, to guide a lightweight, dynamics-aware refiner via rapid online adaptation.

- We introduce VelFlow, a conditional normalizing flow policy that learns the multi-modal distribution of kinematically-sound actions. This approach effectively overcomes the "disastrous averaging" problem inherent in deterministic imitation learning.

- A training strategy that achieves state-of-the-art navigation performance without any costly robot-specific data. Our key innovation is a guided RL phase with curriculum-based annealing of the expert guidance, enabling both stable and rapid adaptation to new embodiments.

## 2 RELATED WORK

### 2.1 CROSS-EMBODIMENT NAVIGATION

Classical local planning methods, such as the Dynamic Window Approach (DWA) (Fox et al., 2002) and Timed Elastic Band (TEB) (Rösmann et al., 2012), have proven robust for local obstacle avoidance. However, their performance is highly sensitive to manual parameter tuning and they are prone to failure in complex, cluttered environments, limiting their generalization. Critically, their core logic provides a strong source of kinematically-aware decisions, a characteristic we leverage for our expert data generation.

Deep learning approaches have diverged. End-to-end (E2E) methods (Wang et al., 2025) map observations directly to joint commands, but deeply entangle planning with dynamics, requiring massive embodiment randomization. Hierarchical methods decouple planning and control. The high-level planner often generates waypoints (Cai et al., 2025; Yang et al., 2023; Shah et al., 2023a; Yang et al., 2024) or velocity commands (Xu et al., 2025b; Liu et al., 2025b; Hirose et al., 2023; Truong et al., 2021) as targets. However, these approaches frequently neglect the embodiment-specific dynamics or tracking errors of the underlying controller, which can result in suboptimal navigation behaviors.

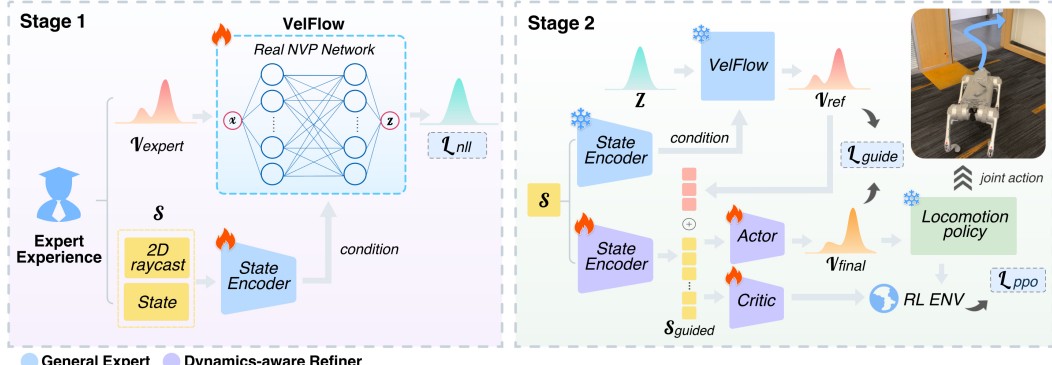

Figure 1: Overview of the CE-Nav two-stage framework. Stage 1 (Left): A multi-modal, embodiment-agnostic General Expert is trained offline via imitation learning on expert data. Stage 2 (Right): The frozen expert is used as a guiding prior to train a Dynamics-Aware Refiner via online reinforcement learning, allowing it to adapt to a specific robot's dynamics.

## 2.2 MODELING MULTI-MODALITY IN ROBOTIC LEARNING

Most deep learning methods for navigation treat the task as a deterministic regression problem (Xu et al., 2025b; Liu et al., 2025c;d). This formulation is ill-suited for scenarios with inherent decision ambiguity (e.g., a T-junction), leading to the well-known "disastrous averaging" issue.

Recognizing this, recent works have explored generative models. Diffusion policies (Chi et al., 2023) and flow models (Papamakarios et al., 2021; Lipman et al., 2022) have shown promise in capturing the full distribution of expert actions, and are utilized in Sridhar et al. (2024); Shah et al. (2023b). However, their application has been largely confined to pure imitation learning contexts.

Adapting these methods to reinforcement learning presents significant challenges. Directly fine-tuning generative policies via RL (Yang et al., 2025; Pfrommer et al., 2025) risks *catastrophic interference*, where aggressive dynamic adaptation gradients override geometric priors. Similarly, conventional demonstration-guided RL (Rajeswaran et al., 2018; Xu et al., 2025a; Smith et al., 2023) often coerce multi-modal demonstrations into a deterministic policy, leading to averaging behaviors in ambiguous scenarios. Meanwhile, curriculum-based methods (Zhao et al., 2022; Nakamoto et al., 2023) typically treat the prior as a "ground truth" to be strictly preserved. Furthermore, standard Residual RL (Ankile et al., 2024) relies on additive architectures which implicitly assume local optimality. This formulation struggles to implement substantial corrections when the required dynamic actions deviate significantly from the reference policy.

Our work differs fundamentally by integrating a flow-based generative model (VelFlow) into a decoupled IL-then-RL framework. Instead of end-to-end fine-tuning or additive residuals, we adopt a "frozen prior" strategy where VelFlow serves as a stable geometric anchor. This unique architecture enables the policy to preserve multi-modal "common sense" while the refiner, via a *conditional refinement* framework, explicitly learns to *deviate* from idealized plans to translate them into feasible controls for specific, unseen robot dynamics.

## 3 METHODOLOGY

### 3.1 OVERVIEW AND PROBLEM FORMULATION

The Cross-Embodiment navigation task requires a mobile robot to navigate from a starting position to a goal in an unknown, cluttered environment. The policy is guided by three types of information: 1) environmental observations, such as a 2D LiDAR scan derived from an onboard depth camera or laser sensor; 2) the robot's proprioceptive state; and 3) the goal position relative to the robot. At each timestep $t$, the policy executes an action $a_t$. The objective is to find a policy $\pi$ that minimizes travel time to the goal while ensuring safety (collision avoidance). Our work investigates a framework for training such policies that can be transferred across a wide range of robotic platforms with varying dynamics and morphologies.

Our system is built upon a hierarchical control architecture that decouples high-level planning ($\pi_{\text{high}}$) from low-level control ($\pi_{\text{low}}$). Given a new robot with its specific locomotion policy $\pi_{\text{low}}$, including all its inherent response characteristics and tracking errors, our goal is to learn a safe and efficient high-level navigation policy $\pi_{\text{high}}$ with minimal training overhead.

## 3.2 Two-Stage Training Paradigm: General Knowledge and Fast Adaptation

We propose a two-stage paradigm for training the high-level policy $\pi_{\text{high}}$ (as illustrated in Fig. 1), which elegantly serves our Cross-Embodiment objective:

**Stage 1: Offline Imitation Learning of an Embodiment-Agnostic General Expert.** The goal of this stage is to learn a universal kinematic expert policy, $\pi_{\text{expert}}$. This policy reasons purely at a geometric and logical level, independent of any specific robot's dynamics. It learns the general principles of navigation—how to perceive pathways and avoid obstacles—making its knowledge inherently embodiment-agnostic. We employ a conditional normalizing flow-based network, VelFlow, to model a continuous distribution of velocities, effectively resolving the "disastrous averaging" problem in multi-modal scenarios.

**Stage 2: Online Reinforcement Learning of a Dynamics-Aware Refiner.** In this stage, guided by the pre-trained General Expert, we use a small amount of interaction between the target robot and its environment to quickly learn its specific dynamic characteristics. The refiner module adapts the general velocity commands from the expert into commands that are dynamically feasible and optimal for the current robot. When transferring to a new robot, we simply freeze the General Expert and train only this lightweight refiner, resulting in a highly efficient and stable adaptation process.

## 3.3 Stage 1: Offline Imitation Learning of the General Expert

The core objective of this stage is to build an embodiment-agnostic navigation brain that masters the universal, high-level principles of planning and obstacle avoidance in complex geometric environments.

### 3.3.1 Expert Experience Construction

To eliminate the high cost and embodiment bias of real-world or physics-based data collection, we generate our expert dataset within a 2D simulation environment. In this simulation, the agent is modeled as a circular rigid body, reducing the planning problem to 2D geometric reasoning, independent of any specific robot's morphology or dynamics.

We synthesize the dataset using the Dynamic Window Approach (DWA) (Fox et al., 2002), a classical local planning algorithm. Crucially, instead of using a robot-specific dynamic model, we configure DWA to operate with a set of **general dynamic constraints**. This config-

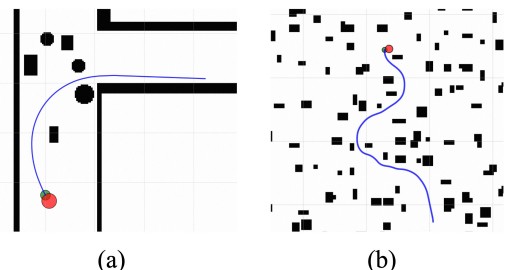

(a)   (b)

Figure 2: Examples of geometry simulation environments used for expert data generation. (a) Corridor environment. (b) Obstacle forest environment.

uration ensures the expert's decisions are not biased towards any specific robot's morphology. The specific parameters are detailed in Table 5.

While classical planners may fail in complex, long-horizon tasks, DWA's core logic provides a robust source of geometrically-sound local decisions. This simplified model ensures the expert focuses purely on geometric collision avoidance, leaving the complex, embodiment-specific dynamics to be learned by the Dynamics-Aware Refiner in Stage 2.

To generate the data, we deployed our DWA planner in tens of thousands of procedurally generated simulation environments with random and complex obstacle layouts (see Fig. 2). To capture navigation's inherent ambiguity and provide rich data for our flow model (VelFlow), we explicitly saved multiple distinct high-scoring actions. Specifically, we collected all candidate actions whose

objective function scores $Score$ were within a $\delta$ threshold of the optimal score $Score_{max}$ (i.e., $Score \geq (1 - \delta) \cdot Score_{max}$, where we set $\delta = 0.1$). Finally, after filtering out trajectories that failed to reach their goal, we compiled a final dataset of 10 million state-action pairs, ensuring our expert data contains only successful and geometrically-sound demonstrations.

### 3.3.2 NETWORK ARCHITECTURE

**State Encoder.** The observation $s$ is composed of a 2D LiDAR scan and a robot state vector. The LiDAR scan represents the distances to surrounding obstacles from the robot's current position. It contains a 360-degree 2D raycast (a LiDAR scan with a maximum range of 4 meters) that samples $N_{ray} = 144$ rays at equal angular intervals on the horizontal plane. The robot state is a 7-dimensional tensor. It comprises vectors expressed in the body frame: the normalized goal direction (3D), current linear velocity (2D), and the current angular velocity (1D). This is augmented by the scalar Euclidean distance to the goal (1D). The state encoder processes the 2D LiDAR scan with a three-layer CNN and concatenates the resulting feature map with the robot state vector. This combined representation is then passed through a two-layer MLP to produce a final 256-dimensional state embedding, which serves as the conditional input to the VelFlow network.

**VelFlow Design.** To fundamentally address the disastrous averaging problem, our goal is to learn the complete conditional probability distribution of the expert's actions, $p(x|s)$, rather than a single deterministic mapping. While Diffusion Policy (Chi et al., 2023) and Flow Matching (Lipman et al., 2022) excel in sample diversity, their reliance on multi-step sampling renders them computationally infeasible for our real-time control application. Conditional Normalizing Flow Models (CNFMs) are powerful deep generative models ideal for this task, as they can accurately model and sample from complex, multi-modal distributions in a single propagation. They provides precise, tractable likelihood estimations, which are crucial for interpretability and stable control. We design our VelFlow module based on the Real-NVP architecture (Dinh et al., 2016), consisting of 12 coupling layers with hidden dimensions of 512. It learns to map a simple base distribution $p_z(z)$ (e.g., a standard Gaussian) to the complex expert velocity distribution $p_x(x|s)$. The training objective is to minimize the negative log-likelihood (NLL) of the expert demonstrations.

$$\mathcal{L}_{NLL} = -\mathbb{E}_{(s,x) \sim \mathcal{D}_{expert}}[\log p(x|s)] \tag{1}$$

Once trained, we can generate diverse and plausible reference velocities, $v_{ref}$, by drawing random samples $z$ from the base distribution and transforming them through the learned VelFlow network: $v_{ref} = f_{VelFlow}(z; s)$.

## 3.4 STAGE 2: ONLINE REINFORCEMENT LEARNING OF THE DYNAMICS-AWARE REFINER

We now introduce the Dynamics-aware Refiner to ground the General Expert's abstract plans in the physical reality of a specific robot. This is achieved through a guided RL process.

### 3.4.1 REINFORCEMENT LEARNING FORMULATION

We formulate the navigation task as a Markov Decision Process (MDP) defined by the tuple $(\mathcal{S}, \mathcal{A}, \mathcal{P}, \mathcal{R}, \gamma)$.

**Observation Space** ($\mathcal{S}$): The initial state representation $s$ is consistent with the imitation learning phase, containing a 360-degree raycast and the 7D robot state. This raycast representation is robust to the sim-to-real gap. For policy learning, the state embedding from the State Encoder is concatenated with the reference velocity $v_{ref}$ provided by the General Expert, forming a guided state $s_{guided}$ which is fed to the actor and critic networks. By conditioning the policy on a specific sampled $v_{ref}$, the refiner's task is thus defined not as replicating the expert's multi-modal distribution, but as learning an optimal, dynamics-aware refinement for that single guiding proposal.

**Action Space** ($\mathcal{A}$): The policy outputs a final velocity command $v_{final}$, which is first predicted as a normalized vector $v_{norm}$ and then scaled by predefined velocity limits $V_{lim}$.

$$v_{final} = V_{lim} \cdot (2 \cdot v_{norm} - 1), \quad v_{norm} \in [0, 1] \tag{2}$$

**Reward Function** ($\mathcal{R}$): Our reward function is structured to encourage efficient, smooth, and safe navigation. Because the refiner policy is trained in a closed loop with the specific $\pi_{low}$, the environmental rewards are generated based on the robot's actual achieved trajectory, not its commanded

velocity. This mechanism inherently forces the refiner to learn a compensatory policy for any systemic latencies or tracking errors within the $\pi_{\text{low}}$. Our reward function is composed of the following components (see A.2 for details): **(1) Efficiency and Goal-Oriented Rewards**: $R_{\text{distance}}$ (rewards progress towards the goal), $R_{\text{checkpoint}}$ (encourages sustained progress), $R_{\text{heading}}$ (rewards velocity aligning with the goal direction), and $R_{\text{goal}}$ (a large bonus for task completion). **(2) Movement Smoothness and Stability Rewards**: Penalties for jerky movements ($P_{\text{linear\_smooth}}$, $P_{\text{yaw\_smooth}}$), excessive body tilt ($P_{\text{stability}}$). **(3) Safety Rewards**: A repulsive potential field based on LiDAR readings ($R_{\text{safety}}$) to encourage keeping a safe distance from obstacles, and a large penalty for collisions ($P_{\text{collision}}$).

### 3.4.2 REFINER DESIGN AND GUIDED TRAINING

During the RL phase, the state observation is processed in two parallel streams: 1) it is fed into the frozen General Expert to generate a reference velocity $v_{\text{ref}}$, and 2) it is passed through the refiner's state encoder. The outputs are then concatenated to form the guided state vector $s_{\text{guided}}$, which serves as the complete input for the refiner's actor and critic networks. We use Proximal Policy Optimization (PPO) (Schulman et al., 2017) to train the refiner policy.

One key innovation is a hybrid loss function that balances imitation and exploration through what we term "Principled Deviation":

$$\mathcal{L}_{\text{guide}} = ||\pi_{\text{refiner}}(s_{\text{guided}}) - scale \cdot v_{\text{ref}}||^2 \tag{3}$$

$$\mathcal{L}_{\text{total}} = \mathcal{L}_{\text{PPO}} + \lambda \cdot \mathcal{L}_{\text{guide}} \tag{4}$$

$\mathcal{L}_{\text{PPO}}$ is the standard PPO objective, driving the refiner to discover behaviors that maximize the cumulative environmental reward. $\mathcal{L}_{\text{guide}}$ is an auxiliary guidance loss, where $v_{\text{ref}}$ is a single velocity command sampled from the frozen $f_{\text{VelFlow}}(z; s)$. The $scale$ term is an auto-computed, embodiment-specific hyperparameter for proportionally scaling $v_{\text{ref}}$ into an acceptable range (see A.3 for details). This mean-squared error term acts as an inductive bias, anchoring the refiner's behavior around the expert's sensible proposals, which ensures learning stability and direction.

The guidance strength, $\lambda$, is not static. We employ a curriculum learning strategy by annealing its value over the course of training: **Initial Phase** (e.g., steps 0-1k, $\lambda = 0.5$): Strong guidance forces the refiner to quickly adopt the expert's fundamental navigation logic. **Mid-Phase** (e.g., steps 1k-5k, $\lambda : 0.5 \rightarrow 0.05$): The guidance weight decays exponentially, granting the refiner more autonomy to explore and fine-tune its actions based on the coupled system dynamics and the reward signal. **Final Phase** (e.g., steps >5k, $\lambda = 0.05$): A weak guidance signal remains, primarily serving as a regularizer to prevent catastrophic forgetting or policy drift. This dynamic balance ensures that any deviation the refiner learns from the expert's command is a principled, data-driven optimization for achieving better performance in the real physical world.

## 4 EXPERIMENTS

### 4.1 EXPERIMENTAL SETUP

**Simulation Environment.** All experiments are conducted within the Isaac Sim physics simulator (NVIDIA). We construct a challenging navigation environment, termed the "obstacle forest," an $l \times l$ area populated with $N_o$ cuboid obstacles of random sizes and positions (see Fig. 3 (a)). For training, we set $N_o = 500$, $l = 40m$ and leverage 1024 parallel environments for efficient data collection and policy updates. During an episode, each robot is spawned at a random location with a distant random goal. An episode terminates if

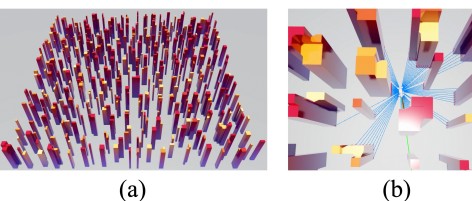

(a)          (b)

Figure 3: (a) The "obstacle forest" with $N_o = 500$, $l = 40m$. (b) Visualization of the 2D raycast input, where blue lines indicate rays (up to a 4m range) that have detected an obstacle.

the robot collides with an obstacle, reaches the goal, or exceeds the maximum episode length. For evaluation, we create four distinct test environments with varying obstacle densities, where $N_o \in \{100, 300, 500, 700\}$ and $l = 20m$. For each difficulty level, we pre-sample and fix 100 start-goal pairs to ensure a consistent and fair comparison across all methods.

Table 1: Ablation study results across four levels of obstacle density.

| Method | Obstacles ($N_o = 100$) | | Obstacles ($N_o = 300$) | | Obstacles ($N_o = 500$) | | Obstacles ($N_o = 700$) | | ETT(h) ↓ |
|---|---|---|---|---|---|---|---|---|---|
| | SR ↑ | SPL ↑ | SR ↑ | SPL ↑ | SR ↑ | SPL ↑ | SR ↑ | SPL ↑ | |
| **CE-Nav (Ours)** | **1.00** | **0.9796** | **0.84** | **0.8001** | **0.83** | **0.7796** | **0.76** | **0.7167** | **6** |
| CE-Nav$_{pure-rl}$ | 0.99 | 0.9452 | 0.64 | 0.6006 | 0.56 | 0.5106 | 0.57 | 0.5179 | 52 |
| CE-Nav$_{regr-rl}$ | 0.46 | 0.4215 | 0.38 | 0.3628 | 0.28 | 0.2666 | 0.35 | 0.3320 | 7 |
| CE-Nav$_{dp-rl}$ | 1.00 | 0.9622 | 0.79 | 0.7499 | 0.77 | 0.7231 | 0.71 | 0.6664 | 52 |
| GE-Only$_{velflow}$ | 0.40 | 0.3675 | 0.01 | 0.0093 | 0.00 | 0.0000 | 0.00 | 0.0000 | N/A |
| GE-Only$_{regr}$ | 0.10 | 0.0909 | 0.00 | 0.0000 | 0.00 | 0.0000 | 0.00 | 0.0000 | N/A |
| CE-Nav$_{\lambda = 0.5}$ | 1.00 | 0.9772 | 0.77 | 0.7409 | 0.73 | 0.7019 | 0.72 | 0.6871 | 6 |

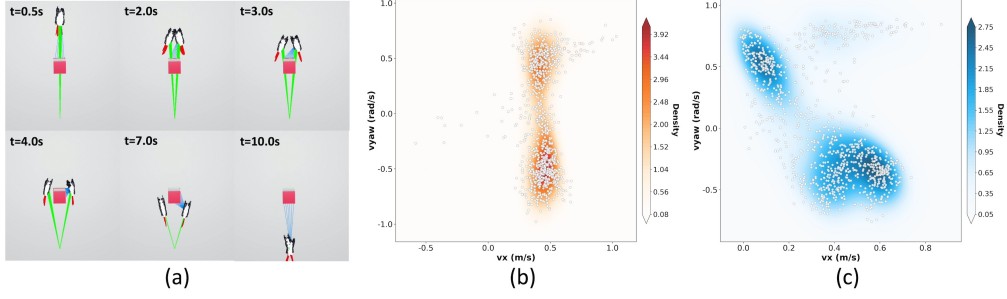

(a)         (b)         (c)

Figure 4: Multi-modal Decision-Making in CE-Nav. (a) 100 robots navigate past an obstacle by splitting into two groups. At the decision point: (b) The expert's reference velocity ($v_{ref}$) proposals form two distinct clusters, representing the choice to turn left or right. (c) The refiner's final velocity commands ($v_{final}$) maintain this bimodal structure while adjusting for dynamics.

**Robot Embodiments.** To assess the Cross-Embodiment generalization capability of our framework, we employ five distinct robot models with radically different dynamics and morphologies: three quadrupeds (Unitree Go2, MagicDog, Spot), one biped (Unitree H1), and one quadrotor (Hummingbird). For quadrotor, we simplify the task to 2.5D navigation by assuming a fixed-altitude controller, allowing them to be commanded using the same 2D velocity interface. The low-level locomotion controllers are sourced from various implementations to represent typical, non-ideal systems with realistic tracking imperfections (see Appendix A.4 for details). This allows us to test our refiner's ability to adapt to controller-specific characteristics.

**Implementation Details.** Our framework consists of two main stages. In the Imitation Learning (IL) stage, the General Expert (GE) is trained offline on the expert dataset using a learning rate of $5 \times 10^{-4}$. After training, its weights are frozen. In the Reinforcement Learning (RL) stage, the learning rates for the actor, critic, and shared feature extractor are set to $5 \times 10^{-4}$, $1 \times 10^{-3}$, and $1 \times 10^{-3}$, respectively. All models are trained and evaluated on a single desktop machine equipped with an NVIDIA RTX 4090 GPU.

**Evaluation Metrics.** We adopt four key quantitative metrics to evaluate performance comprehensively: **Success Rate (SR)**, the percentage of trials where the robot's center of mass reaches within a 0.3-meter radius of the goal without any collisions throughout the episode; **SPL**, the success weighted by the normalized inverse path length for measuring the trajectory efficiency (Anderson et al., 2018); and **Extra Training Time (ETT)**, the wall-clock time required for the additional RL training phase to adapt the policy to a new robot embodiment.

## 4.2 ABLATION STUDIES

We first conduct a series of ablation studies to dissect our framework and validate the necessity and design of its key components. This process establishes the justification for our final proposed model. All ablations are performed on the Unitree Go2.

**The Role of VelFlow and Expert Guidance.** We investigate the impact of our guidance mechanism by comparing our full model against five critical variants: (1) **CE-Nav$_{pure-rl}$**, a pure RL agent

Table 2: Comparisons with other methods on Unitree Go2. The average across all four test environments is reported.

| Method | mSR ↑ | mSPL ↑ | ETT(h) ↓ |
|--------|-------|--------|----------|
| DWA    | 0.6400 | 0.6022 | N/A |
| BC     | 0.0275 | 0.0253 | N/A |
| DP     | 0.0725 | 0.0644 | N/A |
| NavRL  | 0.6925 | 0.6460 | 50 |
| **Ours** | **0.8575** | **0.8190** | **6** |

Table 3: Cross-Embodiment generalization. The average performance across all four test environments is reported.

| Robot Platform | mSR ↑ | mSPL ↑ |
|----------------|-------|--------|
| Unitree Go2    | 0.8575 | 0.8190 |
| Spot           | 0.8325 | 0.7123 |
| MagicDog       | 0.8600 | 0.8231 |
| Unitree H1     | 0.7450 | 0.7223 |
| Hummingbird    | 0.8025 | 0.7491 |

trained from scratch without any expert guidance; (2) **CE-Nav$_{regr-rl}$**, where the VelFlow guidance module is replaced by an MLP regression network with an equivalent number of parameters; (3) **CE-Nav$_{dp-rl}$**, where the VelFlow module is replaced by a diffusion policy model (with an equivalent parameter count) trained on the same expert data; (4) **GE-Only$_{velflow}$**, the General Expert (VelFlow) policy evaluated directly without RL refinement; and (5) **GE-Only$_{regr}$**, the MLP regression-based General Expert policy evaluated directly without RL refinement.

As shown in Table 1, the **GE-Only$_{velflow}$** policy yields a dismal SR, and the **GE-Only$_{regr}$** policy performs even worse. This exposes the classic covariate shift problem in pure IL, and validates that our online refiner is essential for learning a robust recovery policy. The **CE-Nav$_{pure-rl}$** agent confirms the challenge of pure exploration. It requires nearly **9x** the training time of our final model (only 6 hours), and also reaches a significantly lower SR compared to the full model. Most critically, the **CE-Nav$_{regr-rl}$** variant reveals a crucial insight: a suboptimal teacher is more detrimental than no teacher at all. Its SRs are markedly worse than even the pure RL baseline. This is because the regression-based MLP provides an "averaged" and unimodal action prior that fails to capture the multi-modal nature of expert decisions, actively misleading the RL agent. While the **CE-Nav$_{dp-rl}$** variant offers a improvement over CE-Nav$_{regr-RL}$ and CE-Nav$_{pure-RL}$, it presents an undesirable trade-off, as it still falls short of our VelFlow's guide performance and is 8 times more computationally expensive during inference. Visualization in Fig. 4 further illustrates that a high-quality, multi-modal guidance model like VelFlow is the cornerstone of CE-Nav's success.

**Effect of Curriculum-based Guidance Loss.** We validate our curriculum annealing strategy by comparing it against two static weighting schemes: static $\lambda = 0$ (which is **CE-Nav$_{pure-rl}$**) and static $\lambda = 0.5$ (constant strong guidance). Table 1 shows that the **CE-Nav$_{\lambda = 0.5}$** variant is superior to learning without guidance but significantly worse than our curriculum-based approach. While the expert provides a critical starting point, perpetual adherence to its policy stifles exploration. This prevents the agent from discovering more robust or efficient policies that may surpass the expert's own short-sighted behaviors. Our curriculum strategy effectively balances this fundamental imitation-exploration trade-off, leveraging the expert for rapid bootstrapping while gradually empowering the agent to find a superior policy.

## 4.3 COMPARISONS WITH STATE-OF-THE-ART METHODS

We now compare our finalized model against a diverse set of baselines on the Unitree Go2 platform. These include: the **DWA**, a classic local planner for which we carefully tuned dynamics parameters to match the Go2 robot; two IL baselines, **Behavioral Cloning (BC)** (Torabi et al., 2018) and the state-of-the-art **Diffusion Policy (DP)** (Chi et al., 2023); and **NavRL** (Xu et al., 2025b), a state-of-the-art end-to-end RL method for agile navigation using raycast-based observations. Both IL methods were trained on a new complete dataset of 10 million state-action pairs generated from Go2's successful DWA trajectories in the Isaac platform. For a fair comparison, all learning-based baselines were trained in our environment using identical observation and action spaces. Furthermore, NavRL was adapted to train with the same URDF and locomotion policy as our method.

As shown in Table 2, CE-Nav outperforms all baselines. Myopic planners like DWA are ineffective in long-horizon tasks, and IL methods exhibit poor generalization to novel environments. While the strong end-to-end RL baseline, NavRL, performs reasonably, our CE-Nav model surpasses it in

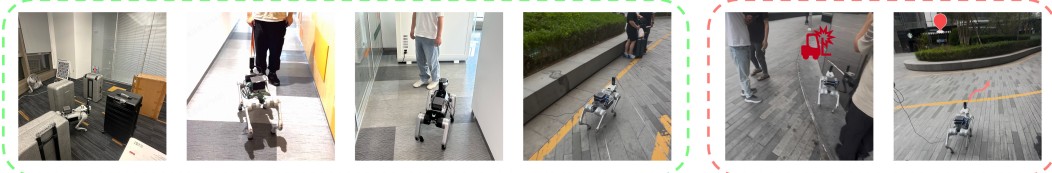

Figure 5: CE-Nav deployment on Unitree Go2 and MagicDog robots. See supp. videos for more cases.

performance while requiring 8x less training time. This substantial gain in both performance and efficiency underscores the effectiveness of our guided, two-stage methodology.

## 4.4 CROSS-EMBODIMENT GENERALIZATION

A key advantage of our proposed method is its ability to transfer to new robot embodiments without requiring any real-world trajectories. We evaluate this by deploying the same pre-trained General Expert across all five robot platforms and running only the brief RL stage.

As shown in Table 3, CE-Nav consistently achieves excellent navigation performance across all five platforms. This demonstrates strong generalization not only across vastly different morphologies and dynamics (e.g., legged vs. aerial), but also across their underlying low-level controllers, which feature a wide range of tracking fidelities (see Appendix A.4). The strong results on the Unitree H1 biped and the Hummingbird quadrotor underscore the powerful adaptation capability of our framework.

## 4.5 REAL-WORLD DEPLOYMENT

It is important to note that our simulated "obstacle forest" is deliberately constructed to be adversarially dense, designed to stress-test the algorithm's robustness in scenarios far exceeding the complexity of typical real-world deployments. We deploy CE-Nav to a Unitree Go2 and a Magic-Dog to validate its sim-to-real transferability (see Appendix A.5 for details). Running on a Jetson Orin NX, our pipeline achieved an inference rate exceeding 10 Hz.

We conducted 40 trials for each of the three challenging scenarios: the indoor obstacle maze, the indoor office corridor, and the outdoor walking path (see Fig. 5). We compared CE-Nav against two baselines: a carefully tuned DWA planner, and the official open-source implementation of NavRL. As shown in Table 4, CE-Nav significantly outperformed both baselines in SR and SPL. DWA frequently failed by becoming permanently trapped in indoor concave regions and corners. While NavRL is designed for generalization (with its original authors claim-

Table 4: Real-world navigation performance comparisons.

| Method | SR ↑ | SPL ↑ |
|---|---|---|
| DWA | 0.7500 | 0.6832 |
| NavRL | 0.5083 | 0.4612 |
| **CE-Nav (Ours)** | **0.9167** | **0.8913** |

ing direct applicability to quadrupeds), its direct deployment on our hardware led to frequent collisions and unstable gaits. This performance gap highlights the critical role of our dynamics-aware refiner, which effectively adapts the general policy to the specific embodiment's physical characteristics, something a purely generalized policy fails to achieve.

**Limitations and Future Work.** CE-Nav's few failures stemmed primarily from sensor limitations rather than planning or control errors. These included collisions with transparent glass walls invisible to LiDAR and detours caused by the absence of RGB perception (see red cases in Fig. 5). This performance underscores the robustness of our core navigation framework and suggests a promising direction for future work: integrating it with advanced perception modules, such as Vision Language Models (VLMs). A prototype of this fast-and-slow system has already been implemented to achieve complex, long-horizon tasks like fetching coffee from Starbucks (**see supp. video**), demonstrating CE-Nav's potential as a pluggable fast system for complex visual navigation tasks.

## 5 CONCLUSION

In this paper, we introduced CE-Nav, a novel framework for Cross-Embodiment local navigation that achieves high performance with remarkable transfer efficiency. By leveraging a hierarchical architecture that decouples high-level planning from low-level control, we successfully isolate the learning of a universal, embodiment-agnostic navigation policy. Our two-stage training paradigm, which combines offline imitation learning of a multi-modal expert with online guided reinforcement learning, proves to be a powerful approach. The VelFlow module effectively addresses the challenge of multi-modal decision-making in navigation, while the dynamics-aware refiner with its curriculum-guided training enables fast and stable adaptation to new robot platforms, by learning a coupled policy that compensates for both the robot's physical dynamics and the execution imperfections of its specific low-level controller. Crucially, our entire framework eliminates the need for collecting expensive and biased real-world robot data. We believe CE-Nav represents a significant step towards truly generalizable and scalable local navigation solutions for the ever-expanding ecosystem of diverse robotic platforms. Furthermore, it provides a robust fast system that can be integrated with high-level planners (such as VLMs), paving the way for next-generation hierarchical navigation systems.

## REPRODUCIBILITY STATEMENT

To ensure the reproducibility of our work, we provide a detailed account of the code, data, experimental setup, and computational requirements. We commit to releasing all code and instructions necessary to replicate our findings upon the acceptance of this paper.

**Code.** Upon acceptance, all code will be made publicly available in a GitHub repository under an MIT license. The repository will include:

- Source code for the **VelFlow** (General Expert) and the **Dynamics-Aware Refiner** models, implemented in PyTorch.
- Scripts for training both Stage 1 (offline IL) and Stage 2 (online RL).
- Code for the procedural generation of simulation environments ("obstacle forest").
- Evaluation scripts to reproduce the results presented in our tables.

**Datasets.** Our framework does not rely on any pre-existing, private datasets. The expert experience for Stage 1 is synthetically generated. The released code will include the complete pipeline for generating this dataset, which consists of 10 million state-action pairs. The expert data is generated using the Dynamic Window Approach (DWA) in a 2D kinematic simulation. All parameters for the DWA planner and the data collection process (as detailed in Section 3.3.1 and Appendix A.1) will be included in the code release, allowing for the exact reconstruction of our dataset.

**Models and Hyperparameters.** The architectures of our models are described in Section 3.3.2. Key hyperparameters for training are provided in Section 4.1. Specifically:

- **VelFlow**: A Real-NVP architecture with 12 coupling layers and hidden dimensions of 512, trained with a learning rate of $5 \times 10^{-4}$.
- **Dynamics-Aware Refiner**: Trained using PPO. The learning rates for the actor, critic, and shared feature extractor are $5 \times 10^{-4}$, $1 \times 10^{-3}$, and $1 \times 10^{-3}$, respectively.
- **Guided Training**: The curriculum for annealing the guidance loss weight $\lambda$ is described in Section 3.4.2 (from 0.5 to 0.05).

All hyperparameters, including the full reward function specification (Appendix A.2, Table 6), will be provided in configuration files within the released code repository to ensure full transparency and ease of replication.

**Experimental Setup.** All simulation experiments were conducted within the Isaac Sim physics simulator. The simulation environments were procedurally generated as described in Section 4.1. For evaluation, we used four fixed test environments with 100 pre-sampled start-goal pairs each to ensure fair and consistent comparisons. The robot embodiments (Unitree Go2, Spot, MagicDog, Unitree H1, Hummingbird) and their respective low-level controllers are detailed in Appendix A.4. The real-world deployment setup is described in Section 4.5 and Appendix A.5.

**Computational Requirements.** The experiments were performed on a single desktop machine equipped with an NVIDIA RTX 4090 GPU. The key computational times are as follows:

- **Stage 1 (Offline IL)**: Training the General Expert on the 10M-pair dataset takes approximately **4 hours**. This is a one-time offline cost, the resulting model is frozen and reused for all embodiments.
- **Stage 2 (Online RL)**: The adaptation of the Dynamics-Aware Refiner for a new robot embodiment takes approximately **6 hours**.

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

# A APPENDIX

## A.1 EXPERT EXPERIENCE CONSTRUCTION PARAMETERS

Table 5: DWA (Dynamic Window Approach) parameters used for generating the embodiment-agnostic expert dataset (Section 3.3.1). These parameters define a set of general dynamic and kinematic constraints for a circular rigid body agent.

| Parameter | Value | Description (Unit) |
|---|---|---|
| *Kinematic Constraints* | | |
| max_speed | 1.5 | Max linear velocity ($m/s$) |
| min_speed | -1.5 | Min linear velocity ($m/s$) |
| max_yaw_rate | 1.57 | Max yaw rate ($rad/s$) |
| min_yaw_rate | -1.57 | Min yaw rate ($rad/s$) |
| *Dynamic Constraints* | | |
| max_accel | 1.5 | Max linear acceleration ($m/s^2$) |
| min_accel | -5.0 | Min linear acceleration (max deceleration) ($m/s^2$) |
| max_delta_yaw_rate | 5.23 | Max yaw angular acceleration ($rad/s^2$) |
| min_delta_yaw_rate | -5.23 | Min yaw angular acceleration ($rad/s^2$) |
| *Planner & Simulation Parameters* | | |
| v_resolution | 0.1 | Velocity sampling resolution ($m/s$) |
| yaw_rate_resolution | 0.17 | Yaw rate sampling resolution ($rad/s$) |
| dt | 0.1 | Simulation time step ($s$) |
| predict_time | 1.0 | Trajectory prediction horizon ($s$) |
| *Robot & Task Parameters* | | |
| robot_radius | 0.2 | Agent's circular radius ($m$) |
| goal_tolerance | 0.4 | Goal tolerance radius (for success) ($m$) |
| *Objective Function Gains (Weights)* | | |
| to_goal_cost_gain | 1.0 | Weight for heading towards the goal |
| speed_cost_gain | 15.0 | Weight for maximizing forward speed |
| obstacle_cost_gain | 0.5 | Weight for distance to obstacles |

## A.2 DETAILED REWARD FUNCTION DESIGN

The function is designed to be dense, consisting of multiple components that guide the agent towards efficient, safe, and stable navigation behaviors (see Table 6). The symbol definitions are:

- $d_t$: The 2D Euclidean distance from the robot to the goal at timestep $t$.
- $\Delta t$: The duration of a single simulation step.
- $v_{\max}$: The maximum configured linear velocity of the robot.
- $\Delta d_{\text{check}}$: The reduction in distance to the goal since the last checkpoint.
- $\hat{\mathbf{h}}$: The robot's current 2D heading unit vector.
- $\hat{\mathbf{g}}$: The 2D unit vector pointing from the robot's current position to the goal.
- $w_{\text{clearance}}$: A safety clearance weight based on forward LiDAR distance, ranging from $[0, 1]$.
- $v_{xy,t}$: The 2D linear velocity vector at timestep $t$.
- $v_{\text{yaw},t}$: The yaw angular velocity at timestep $t$.
- $d_{\text{lidar},i}$: The distance to an obstacle measured by the $i$-th LiDAR ray.
- $\phi, \theta$: The robot's roll and pitch angles.
- $\phi_{\text{th}}, \theta_{\text{th}}$: The safety thresholds for roll and pitch angles.
- $d_0$: The initial 2D distance to the goal at the start of an episode.

Table 6: Reward Function for the online RL stage.

| Reward/Penalty Term | Mathematical Expression | Weight/Description |
|---|---|---|
| *Efficiency & Goal-Orientation* | | |
| $R_{\text{distance}}$ | $\frac{d_{t-1}-d_t}{\Delta t \cdot v_{\max}}$ | +1.0 |
| $R_{\text{checkpoint}}$ | $\Delta d_{\text{check}}$ | +10.0 (calculated every 500 steps) |
| $R_{\text{heading}}$ | $(\hat{\mathbf{h}} \cdot \hat{\mathbf{g}}) \cdot w_{\text{clearance}}$ | +1.0 |
| $R_{\text{goal}}$ | $50.0 \cdot d_0$ | +1.0 (Sparse reward upon episode termination) |
| *Movement Smoothness & Stability* | | |
| $P_{\text{linear\_smooth}}$ | $\|v_{xy,t} - v_{xy,t-1}\|^2$ | -0.5 |
| $P_{\text{yaw\_smooth}}$ | $\|v_{\text{yaw},t} - v_{\text{yaw},t-1}\|^2$ | -0.01 |
| $P_{\text{stability}}$ | $\max(|\phi| - \phi_{\text{th}}, 0)^2 + \max(|\theta| - \theta_{\text{th}}, 0)^2$ | -1.0 |
| *Safety* | | |
| $R_{\text{safety}}$ | $\mathbb{E}_i[\log(d_{\text{lidar},i})]$ | +1.0 |
| $P_{\text{collision}}$ | $50.0$ | -1.0 (Sparse penalty upon episode termination) |

## A.3 DEFINITION OF GUIDANCE LOSS SCALE

The $scale$ parameter used in the guidance loss $\mathcal{L}_{\text{guide}}$ (Equation 3) is automatically computed to safely map the velocity range of the embodiment-agnostic General Expert ($v_{\text{ref}}$) to the specific command range of the target robot. This ensures that the guidance signal $scale \cdot v_{\text{ref}}$ remains within the physical capabilities of the specific hardware.

The calculation is based on two sets of velocity limits:

1. **Expert Limits ($L_{dwa}$):** These are the maximum absolute velocities defined during the DWA expert data generation (see Table 5). The expert's output $v_{\text{ref}}$ is drawn from a distribution learned from this data.
   - $v_{max,dwa}^x = \texttt{max\_speed} = 1.5\,m/s$
   - $v_{max,dwa}^y = \texttt{max\_speed} = 1.5\,m/s$
   - $v_{max,dwa}^{yaw} = \texttt{max\_yaw\_rate} = 1.57\,rad/s$

2. **Embodiment Limits ($L_{emb}$):** These are the specific maximum absolute velocities for the target robot platform (e.g., Go2, A1, etc.). These are defined as part of the robot's hardware configuration and low-level controller $\pi_{\text{low}}$.
   - $v_{max,emb}^x$ (Max forward/backward velocity for the specific robot)
   - $v_{max,emb}^y$ (Max strafing velocity for the specific robot)
   - $v_{max,emb}^{yaw}$ (Max turning velocity for the specific robot)

To find a single, safe scaling factor, we first compute the scaling ratio $scale$ for each axis independently by dividing the embodiment's limit by the expert's limit:

$$scale_x = \frac{v_{max,emb}^x}{v_{max,dwa}^x} \tag{5}$$

$$scale_y = \frac{v_{max,emb}^y}{v_{max,dwa}^y} \tag{6}$$

$$scale_{yaw} = \frac{v_{max,emb}^{yaw}}{v_{max,dwa}^{yaw}} \tag{7}$$

The final $scale$ is then set to the minimum (the most conservative or "safest") of these three ratios. This guarantees that even if the expert proposes a command at its own maximum limit (e.g., $v_{\text{ref}} = (1.5, 0, 0)$), the scaled guidance command ($scale \cdot v_{\text{ref}}$) will not exceed the target robot's maximum velocity on *any* axis.

$$scale = \min(scale_x, scale_y, scale_{yaw}) \tag{8}$$

## A.4 LOCOMOTION CONTROLLER DETAILS

The low-level locomotion controller ($\pi_{\text{low}}$) for each robot platform is sourced from various implementations to represent realistic, non-ideal systems. We train the Unitree Go2 controller using the Isaac Lab framework (Mittal et al., 2023). The Spot quadruped utilizes a publicly available locomotion policy checkpoint[1]. The MagicDog employs a proprietary controller provided by the manufacturer. For the Unitree H1, we use the pre-trained locomotion policy provided in the official Isaac Lab documentation[2]. The Hummingbird quadrotor's controller is from the open-source OmniDrones framework (Xu et al., 2023).

We followed the evaluation method presented by Radosavovic et al. (2024) to quantify the performance and inherent imperfections of these controllers. The results are summarized in Table 7.

Table 7: Details of the low-level locomotion controllers used for each robot platform.

| Robot Platform | Controller Source | Tracking Error (m) ↓ |
|---|---|---|
| Unitree Go2 | In-house (Isaac Lab) | 0.52 |
| Spot | Open-source CKPT[1] | 0.97 |
| MagicDog | Manufacturer-provided CKPT | 0.28 |
| Unitree H1 | Open-source CKPT[2] | 1.56 |
| Hummingbird | Open-source (Omnidrones) | 0.20 |

## A.5 REAL WORLD DEPLOYMENT DETAILS

We deployed the policy on two robots: a Unitree Go2 and a MagicDog. The observation pipeline was adapted to their respective sensors to produce the 2D raycast scan required by our model. For the Go2, the onboard 4D LiDAR's point cloud was first used to generate a 2.5D height map, which was then compressed in Bird's Eye View (BEV) space into a 2D occupancy map. For the MagicDog, the point cloud from its 1D LiDAR (scanning a fixed horizontal plane) was directly converted into a 2D occupancy map. Both of these intermediate occupancy maps were then processed into the final 2D raycast scan input.

## A.6 COMPARISON WITH BARN CHALLENGE SOTA

We compared our method against LiCS-KI, the top-performing learning-based algorithm from the BARN Challenge. Both methods were evaluated on the Jackal robot within our Isaac Sim environment across varying obstacle densities. As shown in Table 8, CE-Nav consistently outperforms LiCS-KI, particularly in dense scenarios ($N_o = 700$), achieving a 23% improvement in success rate. This demonstrates that decoupling kinematic planning from dynamic adaptation yields superior robustness compared to single-stage baselines.

Table 8: Comparison with BARN Challenge SOTA (LiCS-KI) on the Jackal robot.

| Method | $N_o = 100$ | $N_o = 300$ | $N_o = 500$ | $N_o = 700$ |
|---|---|---|---|---|
| **CE-Nav (Ours)** | **1.00** | **0.99** | **0.91** | **0.81** |
| LiCS-KI | 1.00 | 0.97 | 0.71 | 0.58 |
| *Gain* | +0% | +2% | +20% | +23% |

## A.7 ABLATION STUDY ON GENERATIVE MODELS

We investigated the choice of the generative model for the General Expert by comparing our Conditional Normalizing Flow (VelFlow) against Rectified Flow (RF) variants. We evaluated a standard RF (10 sampling steps) and a distilled Reflowed RF (1 sampling step).

---

[1]https://huggingface.co/Kyu3224/quadruped-locomotion-policy
[2]https://docs.isaacsim.omniverse.nvidia.com/latest/robot_simulation/ext_isaacsim_robot_policy_example.html

Table 9 presents the trade-off between inference latency and performance. While Standard RF offers competitive performance, it is approximately $7\times$ slower than our method. Conversely, Reflowed RF achieves extremely low latency but suffers a significant drop in success rate (0.8150 mSR), likely due to distillation errors in capturing the complex multimodal velocity distribution. Our CNF-based approach achieves the highest success rate (0.8575) while maintaining negligible inference latency (0.01 ms), proving to be the optimal choice for real-time control.

Table 9: Ablation study of Generative Models: Normalizing Flow vs. Rectified Flow variants.

| Method | Sampling Steps | Inference Time (ms) | mSR $\uparrow$ |
|---|---|---|---|
| **Ours (VelFlow)** | **1** | **0.01** | **0.8575** |
| Rectified Flow | 10 | 0.07 | 0.8300 |
| Reflowed RF | 1 | 0.007 | 0.8150 |

## A.8 SENSITIVITY TO PLANNER BIAS

To verify whether VelFlow overfits the specific sampling patterns of the DWA planner, we constructed a mixed dataset containing an equal order of magnitude of DWA trajectories and trajectories generated by the TEB (Timed Elastic Band) planner. We then trained a "Mixed-Expert" based model for comparison.

Table 10 compares the Success Rate (SR) of both models across four test environments. The average success rate of the Mixed-Expert based model is almost identical to that of the pure DWA based model (0.8550 vs. 0.8575). This result strongly demonstrates that VelFlow learns the underlying **Universal Geometric Reasoning**—specifically, the ability to identify "traversable space"—rather than overfitting to the algorithmic preferences of a specific planner. Therefore, we retained DWA as the data source in our final design primarily due to its efficiency advantage in large-scale data generation, without sacrificing generalization capabilities.

Table 10: Comparison of VelFlow trained on pure DWA data vs. Mixed (DWA+TEB) data.

| Method | $N_o = 100$ | $N_o = 300$ | $N_o = 500$ | $N_o = 700$ | **mSR** |
|---|---|---|---|---|---|
| **CE-Nav (Ours)** | 1.00 | 0.84 | 0.83 | 0.76 | **0.8575** |
| Mix-Expert (DWA+TEB) | 1.00 | 0.85 | 0.82 | 0.75 | 0.8550 |

## A.9 REAL-TIME CONSTRAINTS AND COMPUTATIONAL ROBUSTNESS

Addressing concerns about the sufficiency of the 10Hz control frequency and potential compute contention, we conducted frequency ablation studies and simulated latency tests. We tested system performance under different control frequencies and with injected random computational delays (simulating compute jitter).

The results are summarized in Table 11. Even when the control frequency is halved to **5Hz**, the average success rate drops only slightly ($< 5\%$). This provides strong evidence that the 10Hz baseline provides ample safety redundancy. Furthermore, under injected random delays of **50ms–100ms** (simulating severe compute contention), the model maintains a high success rate of **0.7625**. This validates that the Stage 2 RL Refiner is robust to system latencies through closed-loop training, eliminating reliance on perfect real-time execution.

Table 11: System performance under different control frequencies and simulated latencies.

| Setting | mSR | Description |
|---|---|---|
| **10Hz (Baseline)** | **0.8025** | Original Setting |
| 5Hz | 0.7650 | Frequency Halved (Simulating Heavy Throttling) |
| 10Hz + Random Delay | 0.7625 | Injected 50ms-100ms Random Delay |

## A.10 Sensitivity to DWA Hyperparameters

To quantify the sensitivity of our method to DWA hyperparameters, we conducted an ablation study focusing on the most critical parameter: **agent radius**. Our original expert data was generated with a radius of 0.2m. We generated two additional expert datasets with significantly different radius settings: a smaller radius (0.1m), simulating an expert that is "overly optimistic" about clearance, and a larger radius (0.3m), simulating an "overly conservative" expert.

We evaluated these models on the Unitree Go2 robot. The results are shown in Table 12. The experimental data demonstrates that our method is **highly insensitive** to DWA hyperparameter settings. Even with substantial variations in the radius parameter ($\pm 50\%$), the fluctuation in final navigation performance is negligible (the difference in mean Success Rate is $< 1\%$). This robustness confirms that the **Stage 2 RL Refiner** can effectively adapt the general guidance to the actual physical constraints of the robot, correcting for potentially inaccurate geometric priors.

Table 12: Sensitivity analysis of the DWA agent radius parameter on Unitree Go2.

| DWA Radius Setting | SR ($N_o = 100$) | SR ($N_o = 300$) | SR ($N_o = 500$) | SR ($N_o = 700$) | Mean SR |
|---|---|---|---|---|---|
| **0.2m (Ours)** | 1.00 | 0.84 | 0.83 | 0.76 | **0.8575** |
| 0.1m (Small) | 1.00 | 0.83 | 0.82 | 0.75 | 0.8500 |
| 0.3m (Large) | 1.00 | 0.84 | 0.81 | 0.76 | 0.8525 |

## A.11 Generalization to Large-Scale Robots (Stress Test)

To validate CE-Nav's generalization capability to "large-scale" robots, we designed a controlled variable experiment. In simulation, we artificially modified the physical collision threshold of the Unitree Go2 to a radius of $R = 1.0$m while keeping its dynamics parameters unchanged. We tested this "giant" robot in the medium density environment ($N_o = 300$).

As shown in Table 13, although the Stage 1 expert was trained based on $R = 0.2$m (and might suggest passing through narrow gaps of only 0.5m), the Stage 2 Refiner achieved a robust success rate of **0.78** under the 1m-radius setting. Compared to the standard robot (0.84), the performance drop is minimal considering the **5x increase** in collision radius relative to the expert's training assumption. This demonstrates that the Refiner successfully learned to adapt to the current dimensional characteristics, effectively overriding the expert's geometrically optimistic suggestions to ensure safety.

Table 13: Performance comparison between Standard Go2 and a modified "Giant" Go2 ($R = 1.0$m).

| Obstacle Count ($N_o$) | 300 (Medium) |
|---|---|
| Standard Go2 | 0.84 |
| **Giant Go2** ($R = 1.0$**m**) | **0.78** |

## A.12 Failure Case Analysis: Agile vs. Dynamics-Limited Robots

To investigate the specific failure modes across different robots, we decomposed the failures into **Timeouts** (indicating safe but slow navigation) and **Collisions** (indicating unsafe navigation). We analyzed two representative platforms: the agile **Unitree Go2** and the dynamics-limited **Unitree H1** (biped).

As shown in Table 14, for the agile Go2, the 0% timeout rate implies high maneuverability; the 14.25% collision rate stems primarily from the extreme geometric complexity of our adversarial evaluation scenarios (high density, dead-ends) rather than control defects. Conversely, for the Unitree H1, despite having a massive tracking error of 1.56m (approximately $3\times$ that of Go2), the collision rate (16.50%) is only marginally higher. The drop in success rate is largely driven by a 9.0% timeout rate. This indicates that the Refiner learned a **conservative strategy**—slowing down or pausing to negotiate complex terrain safely—effectively trading speed for safety to compensate for dynamic limitations.

Table 14: Failure mode decomposition comparing the agile Unitree Go2 and the dynamics-limited Unitree H1.

| Robot Model | Tracking Error | Success Rate | Timeout Rate | Collision Rate |
|---|---|---|---|---|
| Unitree Go2 | 0.52m | 85.75% | **0%** | 14.25% |
| Unitree H1 | 1.56m | 74.50% | **9.0%** | 16.50% |

### A.13 THE USE OF LARGE LANGUAGE MODELS

In the preparation of this manuscript, we utilized the Large Language Model (LLM) Gemini 2.5. The primary application of this tool was for grammar correction and language polishing to improve the clarity and readability of the text. The core scientific contributions, methodologies, and conclusions presented in this paper are our own. We take full responsibility for all content, including any potential errors or inaccuracies.

