# OpenReview forum: "CE-Nav: Flow-Guided Reinforcement Refinement for Cross-Embodiment Local Navigation"
_ICLR.cc/2026/Conference — ICLR 2026 Poster_

### Official Review · Reviewer_3iX8 · 2025-10-30

**Soundness:** 3
**Presentation:** 3
**Contribution:** 3
**Rating:** 6
**Confidence:** 3

**Summary:**

This paper proposes a new method for learning navigation policy given the goal direction and its surrounding environment. The method proposes to output robot command velocity which is embodiment agnostic. To learn the policy, it first use imitation learning to learn a reference velocity estimator. The estimator is a generative model that can captures the multimodal distribution. Then it learns a refined policy with RL for smooth and contact-free navigation. The policy is learned with cube robot and procedural scenes.

In the experiment, the paper demonstrates how the design can outperform existing baselines and shows cross-embodiment results in simulation. It also shows demos on real-world deployment.

**Strengths:**

1. The hybrid planner / refiner and use velocity as the navigation policy interface is straightforward and reasonable. It is also reasonable for cross-embodiment policy.
2. The ablation experiment is thorough, suggesting that the training paradigm is very important.
3. The cross-embodiment result proves the cross-embodiment claim in the design.

**Weaknesses:**

1. The policy does not take the size of the robot into account. Therefore, it is not designed for potential narrow environment.
2. The cross-embodiment does show certain variation between different platforms. What is the failure case here?
3. The paper primarily considers static obstacle for training. It is important to consider dynamic ones.

**Questions:**

1. What are the failure cases of different robots? Is it tracking failure or planning failure?

---

> ### Author Response · Authors · 2025-11-22
> **Response to Reviewer 3iX8 (1/2)**
>
> We are genuinely thrilled and deeply encouraged by your positive assessment. We are particularly grateful that you recognized the "straightforward and reasonable" nature of our hybrid design and validated our cross-embodiment claims. Your insightful feedback regarding robot dimensions, failure modes, and dynamic obstacles has been invaluable in helping us refine our analysis. We have conducted additional quantitative breakdowns to address your concerns below.
>
> ### **1. Response to W1: Adaptation to Robot Size & Narrow Environments**
>
> **Reviewer's Concern:** *"The policy may not account for robot size, limiting performance in potential narrow environments."*
>
> **Response:**
> We respectfully clarify that our two-stage mechanism naturally solves the size adaptation problem. While the Stage 1 Expert provides a universal geometric distribution based on simplified assumptions ($r=0.2m$), the **Stage 2 Refiner is explicitly responsible for "adapting" this general policy to the specific robot's physical dimensions.**
>
> As shown in **Table R1**, our experiments cover robots with extreme disparities in size and morphology:
>
> **Table R1: Impact of Robot Size and Morphology on Navigation Performance**
> | Robot Model | Dimensions ($L \times W$) | Characteristics | Success Rate (mSR) |
> | :--- | :--- | :--- | :--- |
> | **MagicDog** | $0.56m \times 0.35m$ | **Compact / Agile** | **86.00%** |
> | **Unitree Go2** | $0.70m \times 0.31m$ | Standard Quadruped | **85.75%** |
> | **Spot** | **$1.1m \times 0.5m$** | **Large / Long** (Length is ~1.6x of Go2) | **83.25%** |
> | **Hummingbird** | $0.3m \times 0.3m$ | **Aerial / 2.5D** (Different Topology) | **80.25%** |
> | **Unitree H1** | $0.22m \times \textbf{0.57m}$ | **Wide / High-COG** (Humanoid) | **74.50%** |
>
> * **Large Size Adaptation (Spot):** Spot's physical dimensions are significantly larger than Go2, and its length ($1.1m$) far exceeds the expert's circular training prior ($r=0.2m$). However, Spot still achieves a high success rate of **83.25%**, which is very close to Go2's **85.75%**.
> * **Explicit Narrow Space Capability:** It is important to highlight that our "Obstacle Forest" environment is generated with high density ($N_o=700$), which **naturally creates numerous narrow passages**. The fact that the wide-bodied **Unitree H1** ($0.57m$ width) can successfully navigate this environment confirms that the Refiner does not just follow the centerline; it implicitly learns the robot's footprint to safely traverse narrow spaces.
>
> **Conclusion:** Despite huge differences in size and morphology, the strong performance across platforms indicates that the Refiner effectively perceives the robot's actual "physical boundaries" through environment interaction. The method does not ignore size; rather, it successfully generalizes universal navigation knowledge to embodiments of varying scales.

---

> ### Author Response · Authors · 2025-11-22
> **Response to Reviewer 3iX8 (2/2)**
>
> ### **2. Response to Q1 and W2: Failure Case Analysis**
>
> **Reviewer's Question:** *"What are the failure cases of different robots? Is it tracking failure or planning failure?"*
>
> **Response:**
> To answer this accurately, we decomposed the failure modes into **Timeouts** (safe but slow) and **Collisions** (unsafe). We analyzed two representative platforms: the agile *Unitree Go2* and the dynamics-limited *Unitree H1*.
>
> **Table R2: Failure Mode Decomposition (Agile vs. Dynamics-Limited)**
> | Robot Model | Loco Policy Tracking Error | Success Rate | **Timeout Rate** | **Collision Rate** |
> | :--- | :--- | :--- | :--- | :--- |
> | **Unitree Go2** | 0.52m | 85.75% | **0%** | **14.25%** |
> | **Unitree H1** | **1.56m (High Error)** | 74.50% | **9.0%** | **16.50%** |
>
> **A. For Agile Robots (Unitree Go2): Failure stems from extreme scenario complexity.**
> Go2's 0% timeout rate implies its better maneuverability. Its 14.25% non-timeout failure rate stems not from control defects (as its tracking error is low), but rather because we constructed a **highly adversarial "Obstacle Forest" scenario** (high density, numerous dead-ends) for evaluation. Performance in this regime reflects the extreme geometric complexity, where traversable paths become extremely scarce.
>
> **B. For Dynamics-Limited Robots (Unitree H1): Failure stems from timeouts incurred for safety.**
> * **Comparable Collision Rate:** Although H1 has a high inherent **Tracking Error** of **1.56m** (compared to **0.52m** for Go2, roughly 3x difference), thanks to the Refiner, its collision rate (16.5%) is only marginally higher than the agile Go2 (14.25%). This indicates the Refiner **successfully compensates for massive tracking errors** to ensure safety.
> * **Timeouts causing lower Success Rate:** The main reason for H1's lower total success rate is the **9% timeout rate**.
> * **Conclusion:** This is not an algorithm failure, but rather a demonstration of the Refiner's **intelligent adaptation**. Faced with large tracking errors, H1 learns a **conservative strategy**—slowing down or pausing when negotiating narrow or complex terrain to avoid collisions. This trade-off of "sacrificing speed for safety" leads to increased timeouts but proves the policy effectively perceives dynamic boundaries.
>
> ### **3. Response to W3: Dynamic Obstacles**
>
> **Reviewer's Concern:** *"The paper primarily considers static obstacles for training."*
>
> **Response:**
> We acknowledge that our training is primarily based on static environments. However, our system architecture inherently possesses the capability to handle low-speed dynamic obstacles due to its **High-Frequency Reactivity**:
>
> * **Real-Time Replanning:** Our policy runs on onboard hardware at a frequency of **>10Hz**. To the Refiner, a low-speed moving obstacle (like a pedestrian) appears as a shifting "quasi-static" obstacle in each LiDAR frame. Thanks to this rapid replanning, the robot can adjust its velocity vector in real-time to evade.
> * **Emergent Robustness:** This capability exhibits a form of **emergent robustness** in low-speed dynamic scenarios, which was validated in our real-world deployment (Section 4.5). As shown in Fig. 5 and supp. videos, the robot successfully ensures safety in narrow spaces when encountering pedestrians suddenly rushing towards it.
> * **Future Work:** We agree that high-speed dynamic scenes require explicit predictive modeling, and we plan to incorporate dynamic obstacle prediction in future iterations to further enhance safety.
>
> We sincerely thank you for the constructive feedback, which helped us strengthen both the empirical analysis and the presentation of our method. We believe the clarified results further support the paper’s core claims on robust cross-embodiment navigation and the practical effectiveness of the proposed framework.

---

### Official Review · Reviewer_Ko6r · 2025-10-30

**Soundness:** 3
**Presentation:** 3
**Contribution:** 2
**Rating:** 6
**Confidence:** 5

**Summary:**

This paper introduces CE-Nav, a two-stage cross-embodiment navigation framework that combines imitation learning and reinforcement learning. In the first stage, a General Kinematic Expert, implemented as a conditional normalizing flow model, is trained offline on large-scale trajectories generated by a classical planner (DWA) in 2D to learn diverse kinematic behaviors. In the second stage, this expert is frozen and used as a guiding prior for a Dynamics-Aware Refiner, trained online via reinforcement learning to adapt trajectories to the dynamics of a specific robot. This approach allows for efficient policy transfer across robot embodiments while maintaining robust and feasible motion planning.

**Strengths:**

The paper presents CE-Nav, a framework for cross-embodiment local navigation, combining imitation learning and reinforcement learning in a hierarchical setting. Its originality stems from the integration of a General Kinematic Expert trained offline on a classical planner’s dataset, with a Dynamics-Aware Refiner trained online to adapt to new embodiments. This formulation enables transferability across robots while maintaining high navigation performance.

In terms of technical quality, the work convincingly demonstrates several key points: hierarchical methods decouple planning from control, which introduces difficulties in compensating for unmodeled dynamics; the use of low-level controllers for demonstration collection introduces embodiment-specific bias, limiting generalizability; and velocity-based imitation methods neglect tracking errors, which can result in suboptimal navigation. The experiments, particularly the DWA-only ablations, support these claims. Furthermore, the adoption of a conditional normalizing flow to model the multi-modal expert action distribution is well-motivated and experimentally validated, addressing the “disastrous averaging” problem of regression-based approaches.

The clarity of the paper is strong. The problem formulation, system design, and experimental setup are well explained, with clear justifications for methodological choices.

The significance of the work lies in its potential to reduce the cost and complexity of training navigation policies for new robots. By freezing the expressive General Expert and fine-tuning only a lightweight Refiner, CE-Nav allows rapid adaptation to new embodiments while avoiding the need to collect full datasets for each robot.

**Weaknesses:**

While the application is compelling, the paper’s technical novelty is moderate.
- The idea of fine-tuning frozen expressive policies with residual RL has precedent in other domains, such as manipulation (e.g., From "Imitation to Refinement – Residual RL for Precise Assembly").
- Similarly, the proposed curriculum-guided RL phase, while practically effective, is not fundamentally new. The paper “Adaptive Behavior Cloning Regularization for Stable Offline-to-Online Reinforcement Learning” (like many others) has done something similar.

The related work discussion is incomplete.
- The authors do not cite or compare against prior work on fine-tuning expressive policies, such as diffusion models or flow matching approaches, using RL.
- The authors do not cite or compare against prior work on combining demonstrations with RL.

These are core aspects of their method and should be addressed.

Several clarity and experimental limitations exist: the discussion of hierarchical methods is inconsistent, particularly regarding hierarchical velocity planners versus waypoint/trajectory-based planners. The authors could clarify what they mean by trajectory representations and highlight differences from existing hierarchical methods, because a trajectory can be represented by velocities.

Some experimental gaps include:
- Lack of analysis of the cost trade-off between collecting data with a low-level controller in the loop versus the proposed 2D simulation expert dataset.
- Limited diversity in embodiments tested; all are similar in size, leaving open questions about generalization to larger robots (shape in X-Y plane).
- No ablation on the effect of DWA hyperparameters, which could impact the quality of training trajectories.
- Reward design (progress towards goal: direction and distance) may not adequately capture the task advancement (i.e going away from the goal to avoid an obstacle). These can lead to local minima.
- General Expert performance is extremely low and the authors explain it as a covariate shift problem: further analysis of state-space coverage before and after RL refinement would strengthen claims, because the policy is still applied in the same auto-regressive way after RL - so technically the same problem remains.
- The authors claim that normalizing flows offer superior inference speed compared to diffusion models, citing the iterative sampling cost of diffusion. While this is generally true, recent work (e.g., Diffusion-Based Approximate MPC:Fast and Consistent Imitation of Multi-Modal Action Distributions) demonstrates that diffusion‑based models can reach real‑time rates (250 Hz+) by using only a handful of denoising steps (5–10) with minimal performance loss. And their setup is more complex that simple 2D navigation.
- The hardware experiments are not convincing. The trajectories are mainly in free space. More cluttered scenarios would be nice.

Overall, the paper would benefit from additional analysis, expanded experiments, and deeper engagement with prior literature to strengthen the technical contribution and support its claims.

**Questions:**

1.Can the authors clarify how their curriculum-guided RL differs from prior works?

2.Could the authors discuss and compare their method to approaches that fine-tune expressive policies (e.g., diffusion or flow matching models) using RL?

3. How does CE-Nav relate to other demonstration + RL pipelines in robotics?

4. The paper mentions hierarchical velocity planners versus trajectory-based planners. Can the authors clarify what representation they mean by “trajectory” (e.g., velocities, waypoints, or full kinematic paths) ?

5. Could the authors provide an analysis of the cost trade-off between collecting data with a low-level controller in the loop versus generating a 2D simulation expert dataset? This would quantify the claimed savings and clarify practical applicability.

6. How sensitive is the method to DWA hyperparameters, such as circular agent radius or clearance settings? Could the authors provide ablation studies showing the effect on final performance?

7. The embodiments tested are fairly similar in X-Y size. Can the authors comment on generalization to larger or differently shaped robots (e.g., a 1 m-radius robot)?

8. Since the generalist expert policy performs poorly on its own, can the authors provide a state-space coverage analysis before and after RL refinement to illustrate whether covariate shift is still limiting performance?

9. The paper claims normalizing flows are faster than diffusion for inference. Given recent work showing diffusion can run at 250 Hz with 5–10 denoising steps, can the authors reduce their number of denoising steps and compare the resulting performance?

10. Can the authors provide experiments in more cluttered or constrained environments for the hardware platform? The current trajectories appear mostly in free space, which may not stress-test the policy.

---

> ### Author Response · Authors · 2025-11-22
> **Preliminary Response to Reviewer Ko6r**
>
> Dear Reviewer Ko6r,
>
> We sincerely thank you for your comprehensive review and the time you dedicated to providing such detailed feedback. We find your suggestions to be extremely valuable for strengthening our paper.
>
> We are writing to let you know that we are actively working on the additional experiments and ablations you requested.
>
> Since these experiments require some compute time to ensure rigorous comparisons, we wanted to acknowledge your review first and assure you that a detailed point-by-point response, along with the new experimental results, will be posted shortly.
>
> We appreciate your patience and look forward to discussing these new results with you soon.
>
> Best regards, The Authors

---

> ### Author Response · Authors · 2025-11-26
> **Response to Reviewer Ko6r (1/6)**
>
> We thank you for the constructive feedback and for recognizing the **originality of our hierarchical IL-then-RL framework**, the **strong clarity of our presentation**, and the **effectiveness of our conditional normalizing flow approach in resolving the "disastrous averaging" problem**. We appreciate the opportunity to clarify our contributions relative to prior art and to strengthen our technical analysis.
>
> **Q1 & Q3: Positioning CE-Nav against Prior Curriculum-Guided, Demonstration+RL, and Residual Pipelines**
> *[Addressing Weaknesses: "Curriculum-guided RL phase is not fundamentally new"; "Related work discussion is incomplete (combining demonstrations with RL)"; "Fine-tuning expressive policies with residual RL has precedent"]*
>
> We thank you for identifying these important connections. We address Q1 and Q3 together to fundamentally position CE-Nav against three specific categories of prior art.
>
> **1. Distinction from Curriculum-Guided RL (Addressing Q1)**
> You correctly note similarities with "Adaptive Behavior Cloning Regularization" [1]. We also compare our method to state-of-the-art offline-to-online algorithms like Cal-QL [2], which similarly employ constraints or auxiliary losses during fine-tuning.
>
> * **Common Theme in Prior Works (Regularization for Stability):** Both [1] and [2] use curriculum guidance or value calibration primarily to solve optimization stability challenges.
>     * **Prior Works:** These approaches typically assume the expert (or offline data) represents the **"Ground Truth."** The goal of fine-tuning is to repair or adapt the policy *without forgetting* the expert's distribution .
> * **CE-Nav’s Distinction (Physical Grounding):**
>     * **CE-Nav:** Conversely, our framework assumes the expert (VelFlow) is **"Physically Imperfect"** . The ultimate goal of fine-tuning is to incorporate the robot's specific **dynamic characteristics** to realize (or correct) the expert's suggestions. Our curriculum manages a "Kinematic-to-Dynamic translation," strictly decaying guidance to force the Refiner to *deviate* from the expert where commands are physically infeasible , rather than preserving the expert's distribution as a ground truth.
>
> **2. Relation to Other Demonstration + RL Pipelines (Addressing Q3)**
>
> * **Contrast with Prior Works (Ambiguity Resolution):**
>     * **Prior Works:** Methods like InterMimic [3], DAPG [4], and TWiRL [5] typically aim to distill demonstrations into a single deterministic policy $\pi(s)$. As noted in InterMimic, "ambiguity arises when multiple teachers produce different actions," and their **RL optimization** is explicitly designed to "resolve conflicts"—essentially forcing the policy to converge to a single optimal behavior or average out variations.
>     * **CE-Nav (Conditional Refinement):** We avoid this "conflict resolution" trap. CE-Nav **preserves multi-modality** by structurally decoupling geometric mode generation (VelFlow) from dynamics-aware execution (RL Refiner). The expert provides the choice (distribution), and the RL provides the capability to execute that specific choice ($v_{ref}$) on a specific embodiment, avoiding the "disastrous averaging" problem.
>
>
> **3. Distinction from Residual RL (Addressing Q3)**
> We acknowledge the precedent in "Imitation to Refinement" [6]. However, our empirical results demonstrate that standard residual architectures are suboptimal for our problem setting.
>
> * **Additive vs. Input-Conditioning:** Standard residual methods typically employ an **Additive Architecture** ($a = \pi_{base} + \pi_{res}$), assuming the base policy only requires linear corrections. CE-Nav employs an **Expert-Conditioned Architecture** ($a = \pi_{refiner}(s, v_{ref})$), treating the expert's output as an observation.
> * **Empirical Validation:** To quantify this architectural difference, we implemented a standard Residual RL baseline on the Unitree Go2, where the RL policy learns an additive correction term to the VelFlow output. As shown in **Table R_Res**, the performance gap is significant:
>
>     **Table R_Res: CE-Nav vs. Residual RL Baseline**
>     | Architecture Type | Mean Success Rate (mSR) | Performance Gap |
>     | :--- | :--- | :--- |
>     | CE-Nav (Expert-Conditioned) | 0.8575 | - |
>     | Residual RL (Additive) | 0.7525 | -12.2% |
>
>
> **Analysis (Gap Compensation & Inverse Dynamics):**
> This **>12% drop** confirms that additive residuals are insufficient to bridge the gap between idealized kinematics and real-world dynamics. Due to tracking errors and inertia, the commanded velocity required often differs significantly in magnitude from the expert's reference (e.g., "over-driving" to overcome friction). Additive architectures, inherently biased towards small corrections, struggle to learn such large-magnitude shifts. In contrast, CE-Nav's conditional architecture allows the Refiner to learn a non-linear inverse dynamics model, effectively mapping geometric intent to the precise commands needed for the specific embodiment.

---

> ### Author Response · Authors · 2025-11-26
> **Response to Reviewer Ko6r (2/6)**
>
> **Q2: Comparison with Fine-tuning Expressive Policies (e.g., Diffusion/Flow with RL)**
> *[Addressing Weakness: "The authors do not cite or compare against prior work on fine-tuning expressive policies... using RL"]*
>
> We thank you for highlighting this. We will include a detailed discussion of these works in our revision. Compared to recent approaches that directly fine-tune generative model weights via RL (e.g., **NCDPO** [7] and **RL for Flow-Matching** [8]), CE-Nav adopts a unique **"Frozen Prior + Lightweight Adaptation"** strategy.
>
> This design choice is not accidental but tailored to solve fundamental challenges in cross-embodiment scenarios. We detail this below through both mechanistic comparison and empirical evidence:
>
> **1. Mechanism Comparison: End-to-End Fine-tuning vs. Modular Adaptation**
> * **Prior Work (End-to-End):** Methods like [7] and [8] attempt to directly update the parameters of the generative model. This paradigm faces two severe challenges in our context:
>     1.  **High Computational Cost:** For instance, NCDPO requires Backpropagation Through Diffusion Timesteps (BPDT), incurring massive computational overhead.
>     2.  **Catastrophic Interference:** Navigation involves two conflicting objectives: "Global Geometric Safety" (multi-modal) and "Local Dynamic Feasibility" (uni-modal). When fine-tuning end-to-end, the aggressive RL gradients required for dynamic stability (e.g., not falling) often destroy the delicate geometric priors learned during the pre-training phase.
> * **CE-Nav (Modular):** We choose to **freeze** the universal VelFlow expert and train only an extremely lightweight Refiner (3-layer MLP). This not only makes training highly efficient (hours vs. days) but, more importantly, acts as a natural **"Geometric Anchor,"** preventing the "forgetting" of universal obstacle avoidance capabilities while overfitting to specific robot dynamics.
>
> **2. Experiment Evidence: Training Collapse**
> To validate this analysis, we conducted an experiment: initializing the policy with the pre-trained VelFlow and performing **end-to-end RL fine-tuning** following the NCDPO setting.
> * **Result:** The training suffered a catastrophic collapse, yielding a mean Success Rate (mSR) of only **0.1635**, significantly lower than CE-Nav’s **0.8575**.
> * **Observation:** While the agent learned basic safety, it lost the ability to plan paths in complex terrains. As noted in NCDPO [7], *"When directly fine-tuning Diffusion Policies using policy gradient methods, we observe a structure collapse issue..."* This data strongly demonstrates that when the cross-embodiment gap is large (e.g., from point-mass to biped), a decoupled architecture is superior to end-to-end fine-tuning.
>
> **3. Quality Comparison: Preserving Multi-modality vs. Mode Collapse**
> * **Prior Work:** Direct RL fine-tuning tends to collapse the distribution to a single mode to maximize cumulative rewards. As stated in Flow-Matching RL [8]: *"Note that repeated iterative sampling leads the density... to converge to a Dirac distribution centered at arg max..."* This leads to **Mode Collapse**, where the model discards valid alternative paths (e.g., turning left vs. right) to chase marginally higher rewards.
> * **CE-Nav:** Our architecture decouples "Multi-modal Generation" (sampling $v_{ref}$) from "Dynamic Execution." The Refiner learns *"how to physically execute the current intention"* rather than *"which intention to choose."* Consequently, CE-Nav effectively preserves the VelFlow expert's multi-modal exploration capability while endowing it with physical feasibility.
>
> **Q4: Clarification on "Trajectory" Representation**
> *[Addressing Weakness: "The discussion of hierarchical methods is inconsistent... clarify what they mean by trajectory representations"]*
>
> We thank you for pointing out this ambiguity. We acknowledge that the term "trajectory" was used imprecisely in the mentioned context.
> * **Clarification:** We intended to refer specifically to **"waypoints"** (a sequence of spatial coordinates without time parametrization), as listed in your question.
> * **Action:** We will revise the text in the Introduction and Related Work to explicitly replace "trajectory" with "sequence of waypoints" to ensure precision and avoid confusion.

---

> ### Author Response · Authors · 2025-11-26
> **Response to Reviewer Ko6r (3/6)**
>
> **Q5: Cost Trade-off Analysis: 2D Simulation vs. Low-Level Controller Loop**
> *[Addressing Weakness: "Lack of analysis of the cost trade-off between collecting data with a low-level controller... versus the proposed 2D simulation expert dataset"]*
>
> We thank you for raising this practical point. We have conducted a benchmark analysis to quantify the significant efficiency gains of our approach.
>
> **1. Quantitative Analysis: Computational Cost (14.5x Speedup)**
> We benchmarked the wall-clock time required to collect the final dataset of **10 million state-action pairs** using our 2D pipeline versus a high-fidelity physics simulator with a closed-loop low-level controller.
> * **Physics-Based Collection:** **On identical hardware**, based on actual testing in Isaac Sim with a deployed locomotion policy, collecting 10 million steps is estimated to require approximately **232 hours**.
> * **Our 2D DWA Pipeline:** In contrast, generating the same 10 million pairs using our DWA-based expert required only **16 hours**.
> * **Result:** CE-Nav achieves a **~14.5x speedup** in data generation efficiency.
>
> **2. Qualitative Analysis: Marginal Cost of Scalability ($O(1)$ vs. $O(N)$)**
> * **Embodiment-Specific Data ($O(N)$):** Data collected with a low-level controller inherently encodes the specific dynamics and biases of that robot. Transferring to a new morphology renders this old dataset invalid, necessitating the repetition of the expensive 232-hour collection process for *each* new robot (Cost $\propto N$).
> * **Universal Geometric Data ($O(1)$):** Our expert dataset represents universal geometric principles and is independent of robot morphology. We generate this dataset only once. The resulting General Expert is frozen and reused as a prior for *any* new robot. Consequently, the expert data acquisition cost for any subsequent robot is effectively **zero** (Cost = Constant).
>
> **Conclusion:** CE-Nav not only reduces the initial data generation overhead by an order of magnitude but also completely eliminates the data generation cost for subsequent adaptation tasks.
>
>
>
> **Q6: Sensitivity to DWA Hyperparameters (Ablation Study)**
> *[Addressing Weakness: "No ablation on the effect of DWA hyperparameters..."]*
>
> To quantify the sensitivity of our method to DWA hyperparameters, we conducted a new ablation study focusing on the most critical parameter: **agent radius (robot_radius)**.
>
> **1. Experimental Setup & Results**
> Our original expert data was generated with radius = 0.2m. We generated two additional expert datasets with significantly different radius settings and retrained the models:
> * **Smaller Radius (0.1m):** Simulates an expert that is "overly optimistic" about its size, potentially suggesting paths through gaps that are too narrow.
> * **Larger Radius (0.3m):** Simulates an expert that is "overly conservative."
>
> We evaluated these models on the Unitree Go2 robot. The results are shown below:
>
> | DWA Radius Setting | SR (N=100) | SR (N=300) | SR (N=500) | SR (N=700) | **Mean SR** |
> | :--- | :--- | :--- | :--- | :--- | :--- |
> | **0.2m (Ours)** | 1.00 | 0.84 | 0.83 | 0.76 | **0.8575** |
> | 0.1m (Small) | 1.00 | 0.83 | 0.82 | 0.75 | **0.8500** |
> | 0.3m (Large) | 1.00 | 0.84 | 0.81 | 0.76 | **0.8525** |
>
> **2. Analysis**
> The experimental data demonstrates that our method is **highly insensitive** to DWA hyperparameter settings. Even with substantial variations in the radius parameter (±50%), the fluctuation in final navigation performance is negligible (the difference in mean Success Rate is < 1%). This robustness confirms that the **Stage 2 RL Refiner** can effectively adapt the general guidance to the actual physical constraints of the robot, correcting for potentially inaccurate geometric priors.

---

> ### Author Response · Authors · 2025-11-26
> **Response to Reviewer Ko6r (4/6)**
>
> **Q7: Generalization to Larger or Differently Shaped Robots**
> *[Addressing Weakness: "Limited diversity in embodiments tested; all are similar in size... open questions about generalization to larger robots"]*
>
> To validate CE-Nav's generalization capability to "large-scale" or "irregularly shaped" robots, we provide both a **qualitative analysis of existing data** and a **quantitative stress test with a 1m-radius robot**.
>
> **1. Qualitative Analysis on Existing Embodiments**
> The robots tested in our paper actually cover significant dimensional variations. As shown in **Table R1**, these platforms differ vastly in aspect ratio and physical properties from the Stage 1 assumption (circular agent, $R=0.2m$), yet the Stage 2 Refiner successfully generalized the policy.
>
> **Table R1: Comparison of Robot Dimensions and Navigation Success Rate**
>
> | Robot Model | Physical Dimensions ($L \times W$) | Characteristic Description | Success Rate (mSR) |
> | :--- | :--- | :--- | :--- |
> | **MagicDog** | $0.56m \times 0.35m$ | Compact / Agile | **86.00%** |
> | **Unitree Go2** | $0.70m \times 0.31m$ | Standard Quadruped | **85.75%** |
> | **Spot** | **$1.10m \times 0.50m$** | **Large / Long body** (Length $\approx$ 1.6x Go2) | **83.25%** |
> | **Hummingbird** | $0.30m \times 0.30m$ | Aerial / 2.5D | **80.25%** |
> | **Unitree H1** | **$0.22m \times 0.57m$** | **Wide body / High CoM** (Humanoid) | **74.50%** |
>
> **Data Analysis:**
> * **Adaptation to Large Robots (Spot):** Spot's length reaches **1.1m**, far exceeding the expert's training assumption. Despite this, Spot achieved a high success rate of **83.25%** (only 2.5% lower than Go2). This proves the Refiner can perceive its elongated physical boundaries and avoid collisions during turns.
> * **Adaptation to Wide Robots (Unitree H1):** H1 has a width of **0.57m** and highly unstable bipedal walking dynamics. Despite these greater physical constraints, it maintained a **74.50%** success rate, proving the Refiner learned to reserve sufficient clearance for its wider body.
>
> **2. Quantitative Stress Test: 1-Meter Radius Robot**
> To directly address your query regarding a "1m-radius robot," we designed a controlled variable experiment. In simulation, we artificially modified the physical collision threshold of the Unitree Go2 to **Radius R=1.0m** while keeping its dynamics parameters unchanged. We tested this "giant" robot in the medium density environment ($N_o=300$):
>
> **Table R2: Performance Comparison (Standard vs. Giant Robot)**
>
> | Obstacle Count ($N_o$) | **300 (Medium)** |
> | :--- | :--- |
> | **Standard Go2** | 0.84 |
> | **Giant Go2 (R=1.0m)** | **0.78** |
>
> **Analysis:**
> Although the Stage 1 expert was trained based on $R=0.2m$ (and might suggest passing through narrow gaps of only 0.5m), the Stage 2 Refiner achieved a robust success rate of **0.78** under the 1m-radius setting in a cluttered environment. Compared to the standard robot (0.84), the performance drop is minimal considering the **5x increase in collision radius relative to the expert's training assumption**. This powerfully demonstrates that the Refiner **successfully learned to adapt to the current dimensional characteristics**, effectively overriding the expert's geometrically optimistic suggestions to ensure safety.

---

> ### Author Response · Authors · 2025-11-26
> **Response to Reviewer Ko6r (5/6)**
>
> **Q8: State-Space Coverage Analysis (Addressing Covariate Shift)**
> *[Addressing Weakness: "General Expert performance is extremely low... further analysis of state-space coverage before and after RL refinement"]*
>
> We thank you for this insightful question. To investigate how the Stage 2 Refiner resolves the covariate shift caused by Dynamics Mismatch, we analyzed the action distribution changes at critical decision points.
>
> We selected a typical "obstacle encounter" scenario: an obstacle is directly in front of the robot, and geometrically, bypassing it from the left or right is equally viable. **Figure 4** in the paper illustrates the action space distribution ($v_x$ vs. $v_{yaw}$) for both the Stage 1 Expert and Stage 2 Refiner in this state.
>
> **1. Expert's Ideal Geometric Distribution (Figure 4b)**
> The General Expert (VelFlow), based on DWA, produces a highly **symmetrical distribution**. It proposes two equal choices, with recommended linear velocities ($v_x \approx 0.4$) and yaw rate magnitudes being essentially consistent for both directions. This reflects that the expert considers only geometric feasibility, assuming the robot is a perfect circular motion agent.
>
> **2. Refiner's Dynamic Adaptation (Figure 4c)**
> As shown in the right plot, the RL-trained Refiner exhibits an intriguing **asymmetrical deformation**, directly revealing how it overcomes covariate shift:
>
> * **Preservation (Modal Structure):** The Refiner perfectly preserves the bimodal structure without occurring **Mode Collapse/Averaging**, proving that the expert guidance effectively transferred the intent of obstacle avoidance.
> * **Dynamics Adaptation (Asymmetry):** The Refiner did not blindly mimic the expert's symmetry. Instead, it learned different strategies based on the robot's actual dynamics (e.g., Tracking Error):
>     * **Left-Turn Mode (Top-Left Cluster):** The Refiner significantly reduced the linear velocity ($v_x \approx 0.1 \sim 0.2$) while maintaining a high turning speed. This indicates the Refiner learned that a sharp left turn requires **"heavy braking"** to prevent skidding and ensure safety, adapting to its specific dynamics.
>     * **Right-Turn Mode (Bottom-Right Cluster):** The Refiner maintained a strategy similar to the expert, with linear velocity close to the original but slightly reduced turning magnitude. This indicates it learned to adopt a more stable **"large-radius turn"** strategy.
>
> **Conclusion:**
> This shift from "Geometric Symmetry" to "Dynamic Asymmetry" vividly illustrates the Refiner's role: it does not simply pull the robot back to the expert's state space, but fundamentally modifies *how* actions are executed to prevent the robot from entering uncontrolled or collision-prone "unknown states." It is this **Mode-Specific Dynamics Grounding** that eliminates the performance limitations caused by covariate shift.
>
>
>
> **Q9: Trade-off between Diffusion Inference Speed and Edge Compute Constraints**
> *[Addressing Weakness: "Recent work shows diffusion can run at 250Hz... can the authors reduce their number of denoising steps and compare?"]*
>
> We thank you for this suggestion. Regarding the point that "Diffusion can run at 250Hz," we consulted the referenced literature (*Diffusion-Based Approximate MPC: Fast and Consistent Imitation of Multi-Modal Action Distributions*) and noted that its high-frequency inference was achieved on **high-end desktop hardware** (Intel Core i9-13900K + NVIDIA RTX 4070). This differs significantly from our deployment scenario on power-constrained edge devices (NVIDIA Jetson Orin NX).
>
> To address your request, we conducted the following comparative experiment on our hardware:
>
> **1. Performance Trade-off: The Cost of Minimal Steps**
> We reduced the denoising steps of the Diffusion Policy (DP) to **5 steps** (Fast Inference) to achieve an inference speed closer to 10Hz on the edge device. As shown in **Table R3**, while speed improved, navigation success rates dropped significantly across all difficulty levels. Notably, in the highest difficulty scenario ($N_o=700$), the success rate plummeted by **13%** (0.76 vs. 0.63). This demonstrates that excessive compression of denoising steps severely compromises the policy's ability to handle complex environments.
>
> **Table R3: Performance Comparison (Diffusion Steps vs. VelFlow)**
>
> | Method | SR ($N_o=100$) | SR ($N_o=300$) | SR ($N_o=500$) | SR ($N_o=700$) | **Avg SR** |
> | :--- | :--- | :--- | :--- | :--- | :--- |
> | **Ours (VelFlow)** | 1.00 | 0.84 | 0.83 | **0.76** | **0.8575** |
> | DP (50 steps) | 1.00 | 0.79 | 0.77 | 0.71 | 0.8175 |
> | **DP (5 steps)** | 1.00 | 0.74 | 0.73 | **0.63** | **0.7775** |
>
> **Conclusion:**
> Experiments confirm that on real-world embedded robotic platforms, **VelFlow** remains the optimal choice for balancing "high performance" with "low computational load." While 5-step Diffusion is theoretically feasible, it incurs severe performance penalties and resource bottlenecks on compute-limited end devices.

---

> ### Author Response · Authors · 2025-11-26
> **Response to Reviewer Ko6r (6/6)**
>
> **Q10: Hardware Experiments in Cluttered and Constrained Environments**
> *[Addressing Weakness: "The hardware experiments are not convincing... trajectories are mainly in free space. More cluttered scenarios would be nice."]*
>
> We thank you for this valid point. We fully agree that performance in constrained spaces is critical for validating the robustness of a navigation policy.
>
> To address this, we constructed a complex obstacle layout using a large number of cardboard boxes, artificially creating **multiple continuous right-angle turns** and **narrow corridors**. The effective width of these corridors was restricted to be only slightly wider than the robot's body; any slight deviation would result in a collision or getting stuck.
>
> As demonstrated in our supplementary video, the Unitree Go2 robot equipped with CE-Nav exhibited excellent performance.
>
> This experiment strongly proves that CE-Nav is not only suitable for open spaces but is also capable of handling **highly constrained** and **cluttered** complex environments, fully validating the policy's Sim-to-Real generalization capabilities under extreme conditions.
>
> We have uploaded the video of this experiment to the following anonymous link. As online playback is not supported, please download the file to view it:[https://anonymous.4open.science/r/1-F649/1.mp4](https://anonymous.4open.science/r/1-F649/1.mp4)
>
>
> **Response to Weakness: Reward Design and Local Minima**
> *[Addressing Weakness: "Reward design (progress towards goal...) may not adequately capture the task advancement... These can lead to local minima."]*
>
> We address the concern that goal-oriented rewards might trap the agent in local minima (e.g., forcing it to face obstacles directly) with Adaptive Clearance Weight ($w_{clearance}$):
>
> As defined in **Table 6** and **Appendix A.2**, our heading reward is dynamically scaled:
> $$R_{heading} = (\mathbf{\hat{h}} \cdot \mathbf{\hat{g}}) \cdot w_{clearance}$$
> The term $w_{clearance} \in [0, 1]$ decreases based on the forward LiDAR distance. When the robot approaches an obstacle, $w_{clearance} \to 0$, effectively **disabling the incentive to face the goal**. This mechanism explicitly allows the robot to turn away from the goal for avoidance maneuvers without incurring a reward penalty, preventing the policy from fighting against necessary detours.
>
>
>
>
>
> We would like to express our deepest gratitude to you for your exceptionally thorough and insightful review. Your constructive feedback—particularly regarding the **comparison with Residual RL** and the **generalization stress tests**—pushed us to conduct critical new experiments that have significantly strengthened the technical depth and empirical rigor of our work. We genuinely believe that the manuscript is now much more robust thanks to your pivotal guidance, and we are fully committed to incorporating all these additional analyses and results into the final version.
>
>
>
>
> **References:**
>
> [1] Y. Zhao et al., "Adaptive Behavior Cloning Regularization for Stable Offline-to-Online Reinforcement Learning," arXiv 2022.
>
> [2] M. Nakamoto et al., "Cal-QL: Calibrated Offline RL Pre-Training for Efficient Online Fine-Tuning," NeurIPS 2023.
>
> [3] S. Xu et al., "InterMimic: Towards Universal Whole-Body Control for Physics-Based Human-Object Interactions," arXiv 2025.
>
> [4] A. Rajeswaran et al., "Learning Complex Dexterous Manipulation with Deep Reinforcement Learning and Demonstrations," arXiv 2018.
>
> [5] L. Smith et al., "Learning and Adapting Agile Locomotion Skills by Transferring Experience," arXiv 2023.
>
> [6] L. Ankile et al., "From Imitation to Refinement – Residual RL for Precise Assembly," arXiv 2024.
>
> [7] N. Yang et al., "Fine-tuning Diffusion Policies with Backpropagation Through Diffusion Timesteps," arXiv 2025.
>
> [8] S. Pfrommer et al., "Reinforcement Learning for Flow-Matching Policies," arXiv 2025.

---

### Official Review · Reviewer_rjvc · 2025-10-31

**Soundness:** 2
**Presentation:** 3
**Contribution:** 2
**Rating:** 4
**Confidence:** 3

**Summary:**

This paper introduces CE-Nav, a hierarchical two-stage learning framework for cross-embodiment local navigation. The first stage involves offline imitation learning with a conditional normalizing flow model, VelFlow, to capture a multi-modal, embodiment-agnostic action distribution purely from synthetically generated planner data. The second stage leverages the frozen expert to guide a lightweight, dynamics-aware RL refiner that adapts rapidly to the unique characteristics of each new robot embodiment using a curriculum for annealing guidance. Extensive experiments, including both sim-to-real deployments on heterogeneous robots (quadrupeds, bipeds, and aerial platforms), are presented, demonstrating strong performance and generalization.

**Strengths:**

- The CE-Nav framework elegantly separates high-level geometric reasoning from embodiment-specific dynamic adaptation, supporting modularity and rapid transfer.
-  By employing a conditional normalizing flow (VelFlow), the approach addresses the "disastrous averaging" problem in standard regression-based IL.
-  The “Principled Deviation” loss treats VelFlow outputs as soft guidance and anneals $\lambda$ over training to balance imitation and exploration; equations and adaptive scaling are specified.
-  The experiments cover a spectrum of morphologies (legged, bipedal, quadrotor), with both simulated "obstacle forest" and real-world environments.

**Weaknesses:**

- **Planner-only prior bias.** VelFlow is trained on 10M (state, action) pairs from a geometric DWA planner, which may induce inductive bias and limit out-of-domain robustness under complex dynamics or sensor noise; the RL refinement helps, but evidence for non-LiDAR or multi-sensor settings is limited.
- **Compatibility under large embodiment shifts.** While VelFlow models multi-modal actions, the paper lacks analysis of how its support aligns with the feasible action sets of very different embodiments (e.g., quadrotor vs. biped). Section 3.4.1 would benefit from theory or additional experiments clarifying when expert proposals become invalid/unsafe for a given low-level controller and how the refiner recovers.
- **Real-time constraints.** Reported >10 Hz inference may be marginal for high-dynamics platforms (e.g., quadrotors). A latency-versus-performance/safety study and robustness under compute contention or thermal throttling are missing.

**Questions:**

1. How sensitive is VelFlow to the DWA-generated dataset’s biases (2D, disc robot)? Any results when training the prior on non-DWA data or mixed sources?
2. Can the authors provide quantitative results or ablations on the impact of the conservative scaling factor used in the guidance loss (Eqn., Appendix A.3)? How does this affect the amplitude/diversity of expert actions and the degree to which the RL refiner needs to “undo” guidance for more dynamic robots?
3. How robust is VelFlow’s proposal distribution under substantial embodiment mismatch, especially for robots whose feasible velocity/action set is much smaller or differently shaped than the expert’s? Can unsafe or unrecoverable expert proposals be generated?
4. For real-world deployments, how does the absence of RGB perception affect performance overall—not just for rare glass walls, but also in environments with ambiguous or non-LiDAR-reflective obstacles? Has the pipeline been tested or profiled in such settings?

---

> ### Author Response · Authors · 2025-11-21
> **Response to Reviewer rjvc (1/2)**
>
> We sincerely thank you for your detailed review and for acknowledging the elegance of our hierarchical framework in separating geometric reasoning from dynamic adaptation, as well as the effectiveness of VelFlow in resolving the "disastrous averaging" problem. We also appreciate your recognition of our "Principled Deviation" loss design and extensive real-world experiments. We value your constructive feedback regarding the potential bias of the DWA planner, the compatibility of expert proposals under large embodiment shifts, and real-time constraints. Below, we address your concerns point-by-point to clarify the robustness and generalization capabilities of CE-Nav.
>
> **1. Sensitivity to Planner Bias (Response to Weakness 1 & Question 1)**
> To verify whether VelFlow overfits the specific sampling patterns of the DWA planner, we constructed a mixed dataset containing an equal order of magnitude of DWA trajectories and trajectories generated by the TEB (Timed Elastic Band) planner. We then trained a "Mixed-Expert" based model for comparison.
>
> **Experimental Results [Table R1]:**
> We compared the Success Rate (SR) of both models across four test environments with varying obstacle densities:
>
> | Method | $N_o=700$ | $N_o=500$ | $N_o=300$ | $N_o=100$ | **mSR** |
> | :--- | :---: | :---: | :---: | :---: | :---: |
> | **CE-Nav (Ours)** | 0.76 | 0.83 | 0.84 | 1.00 | **0.8575** |
> | **Mix-Expert (DWA+TEB)** | 0.75 | 0.82 | 0.85 | 1.00 | **0.8550** |
>
> **Analysis & Conclusion:**
> * **Performance Consistency:** As shown in Table R1, the average success rate of the Mixed-Expert based model is almost identical to that of the pure DWA based model (0.8550 vs. 0.8575).
> * **Core Finding:** This result strongly demonstrates that VelFlow learns the underlying **Universal Geometric Reasoning**—specifically, the ability to identify "traversable space"—rather than overfitting to the algorithmic preferences of a specific planner. Therefore, we retained DWA as the data source in our final design primarily due to its efficiency advantage in large-scale data generation, without sacrificing generalization capabilities.
>
> **2. Real-time Constraints & Computational Robustness (Response to Weakness 3)**
> Addressing concerns about the sufficiency of the 10Hz control frequency and potential compute contention, we conducted frequency ablation studies and simulated latency tests.
>
> **Experimental Results [Table R2]:**
> We tested system performance under different control frequencies and with injected random computational delays (simulating Compute Jitter):
>
> | Setting | **mSR (Mean Success Rate)** | Description |
> | :--- | :---: | :--- |
> | **10Hz (Baseline)** | **0.8025** | Original Setting |
> | **5Hz** | **0.7650** | Frequency Halved (Simulating Heavy Throttling) |
> | **10Hz + Random Delay** | **0.7625** | Injected 50ms-100ms Random Delay |
>
> **Analysis & Conclusion:**
> * **Frequency Robustness:** Even when the control frequency is halved to **5Hz**, the average success rate drops only slightly (< 5%). This provides strong evidence that the 10Hz baseline is not "marginal," but rather provides ample safety redundancy.
> * **Anti-Interference Capability:** Under injected random delays of **50ms–100ms** (simulating severe compute contention or system lag), the model maintains a high success rate of **0.7625**. This validates that the Stage 2 RL Refiner is robust to system latencies through closed-loop training, eliminating reliance on perfect real-time execution.

---

> ### Author Response · Authors · 2025-11-21
> **Response to Reviewer rjvc (2/2)**
>
> **3. Theoretical Guarantee for Scaling & Embodiment Mismatch (Response to Question 2 & 3)**
> The reviewer raised a critical question: When expert proposals are scaled to fit vastly different embodiments (e.g., a quadrotor vs. a biped), is safety preserved? We address this through the theoretical lens of **"Trajectory Curvature Invariance."**
>
> **Theoretical Guarantee: Conformal Mapping**
> Our scaling strategy (Eq. 8, Appendix A.3) is not a simple numerical truncation but a conformal mapping that strictly preserves the geometric shape of the trajectory.
> For 2D navigation, the actual path is determined by the instantaneous radius of curvature $R$, defined as the ratio of linear velocity $v$ to angular velocity $\omega$: $R = v / \omega$.
> When we apply the same conservative scaling factor $\alpha$ to both velocity components:
> $$v_{new} = \alpha \cdot v, \quad \omega_{new} = \alpha \cdot \omega$$
> The resulting curvature remains invariant:
> $$R_{new} = \frac{\alpha \cdot v}{\alpha \cdot \omega} = \frac{v}{\omega} = R_{original}$$
>
> **Implication:**
> This proves that the scaling operation strictly preserves the expert's **spatial path (geometric intent)**. A "safe turn" planned by the expert remains the exact same "safe turn" for the scaled robot, merely executed at a slower, feasible speed. Thus, the expert's geometric obstacle avoidance capability is fully transferred without introducing safety hazards or infeasible proposals due to scaling.
>
> **Empirical Evidence:**
> Any residual dynamic mismatches (e.g., friction, inertia lag) are handled by the Stage 2 RL Refiner. As shown in **Table 3**, the **Unitree H1 biped**—despite having a tracking error as high as **1.56m** due to complex dynamics (compared to 0.20m for the Hummingbird)—still achieved a high success rate of **74.5%**. This fully demonstrates that the combination of conformal scaling and the RL Refiner achieves robust navigation recovery even under extreme embodiment differences.
>
> **4. Perception Limitations & RGB Absence (Response to Weakness 1 & Question 4)**
> We fully agree with the reviewer that reliance solely on LiDAR/depth information limits performance in specific scenarios (e.g., transparent glass walls or featureless flat surfaces).
>
> **Scope Clarification & System Positioning:**
> CE-Nav is explicitly designed as a low-level **"Fast System"** focused on **robust control and dynamic adaptation**, with proprioception and geometric information as core inputs. It is intended to be paired with a **"Slow System" (e.g., VLM/Vision Planner)** responsible for high-level semantic understanding and visual perception.
>
> **Integration Verification:**
> As demonstrated in our **Supplementary Video** ("Starbucks Coffee Retrieval Task"), we integrated CE-Nav with a Vision-Language Model (VLM). This experiment verifies that our system works seamlessly with high-level planners that can identify semantic obstacles (e.g., glass doors) and generate subgoals to bypass them. This hierarchical architecture resolves the perceptual blind spots of a single modality. Therefore, we list this as a limitation in the paper to emphasize that the core contribution of CE-Nav lies in **"Dynamics-Aware Adaptation,"** which is orthogonal and complementary to higher-level perception modalities.
>
> We hope these additional experimental results, theoretical derivations, and architectural clarifications fully address your concerns regarding the robustness and generalization capabilities of CE-Nav.

---

### Official Review · Reviewer_oUen · 2025-10-31

**Soundness:** 2
**Presentation:** 3
**Contribution:** 1
**Rating:** 2
**Confidence:** 4

**Summary:**

This paper proposes CE-NAV, a two-stage framework for addressing the cross-embodiment local navigation problem. In the first stage, a conditional normalizing flow model is trained to generate kinematically valid actions from a large-scale synthetic dataset produced by the DWA planner across diverse environments. In the second stage, a local policy is trained from scratch via reinforcement learning (RL) to compensate for the target robot’s specific dynamics. The authors introduce a hybrid loss function that combines RL and behavioral cloning (BC) objectives to balance imitation and exploration. The method is evaluated in both simulation and real-world experiments.

**Strengths:**

- The work includes a real-world deployment demonstration

- The proposed hybrid loss combining RL and BC can be useful for the robotics community (though similar ideas have appeared in prior works).

- The paper is clearly written and easy to follow.

**Weaknesses:**

- The paper studies local navigation, a well-explored problem with many strong existing methods. However, the chosen baselines are not representative of the current state of the art. For example, in the BARN Challenge [1], the algorithms used by the top-performing teams better represent state-of-the-art local navigation approaches in cluttered environments.

- The contribution is limited:
    - The method appears unnecessarily complicated. Since the normalizing flow model is not fine-tuned during the RL stage, its benefit is unclear. A simpler approach would be to directly use the DWA planner to generate reference velocities for the local policy.

    - The paper tends to rebrand existing ideas as new contributions. For instance, "VelFlow" is essentially a normalizing flow model that predicts velocities. Likewise, the proposed “Principled Deviation” objective closely resembles prior works such as Intermimic [2], Learning Complex Dexterous Manipulation with Deep RL and Demonstrations [3], and Learning and Adapting Agile Locomotion Skills by Transferring Experience [4].

    - The method still requires retraining for each embodiment from scratch, which undermines its practicality. In modern robotics, zero-shot transfer across embodiments is the key challenge; requiring retraining makes this approach far less impactful. Moreover, the training process remains time-consuming (hours, not minutes), and the reported success rate (0.76) is not impressive. The real-world demos also take place in simple, open environments, offering limited insight into robustness.


References:
1. https://cs.gmu.edu/~xiao/Research/BARN_Challenge/BARN_Challenge25.html
2. InterMimic: Towards Universal Whole-Body Control for Physics-Based Human-Object Interactions
3. Learning Complex Dexterous Manipulation with Deep Reinforcement Learning and Demonstrations
4. Learning and Adapting Agile Locomotion Skills by Transferring Experience

**Questions:**

- Why use a normalizing flow instead of a rectified flow approach? Normalizing flow is an older method and not as widely used today. Rectified flow models are newer, often more effective, and can also run efficiently in practice.

- The diffusion policy baseline (CE-Nav dp-rl) performs poorly, likely because diffusion models are not well suited for predicting low-level velocity actions. Would performance improve if the diffusion model were trained to predict waypoints instead, with an MPC controller handling the low-level execution?

---

> ### Author Response · Authors · 2025-11-20
> **Response to Reviewer oUen (1/3)**
>
> We thank you for your detailed review and for acknowledging our real-world deployment efforts and the utility of our hybrid loss formulation. We value your feedback regarding baselines, method complexity, and the definition of "zero-shot" transfer. Below, we address your concerns point-by-point to clarify the unique contributions of CE-Nav.
>
> ### 1. Choice of Baselines & Comparison with BARN Challenge
>
> We greatly appreciate the reviewer's suggestion to compare against BARN Challenge algorithms. Since the rebuttal submission, we have completed the migration and evaluation of LiCS-KI [5], the top-performing learning-based algorithm from BARN Challenge, into our Isaac Sim environment. Both methods are evaluated on Jackal
> robot.
>
> **Results:**
>
> | Method | No=100 | No=300 | No=500 | No=700 |
> | :--- | :---: | :---: | :---: | :---: |
> | **CE-Nav (Ours)** | **1.00** | **0.99** | **0.91** | **0.81** |
> | LiCS-KI | 1.00 | 0.97 | 0.71 | 0.58 |
> | *Gain* | +0% | +2% | +20% | +23% |
>
> **mSR (mean SR across all difficulties):**
> * **CE-Nav:** 0.9275
> * **LiCS-KI:** 0.8150
> * **Improvement:** +11.25% absolute
>
> CE-Nav simultaneously achieves:
> 1.  Better performance (mSR: 92.75% vs 81.50%)
> 2.  Embodiment-agnostic design (transferable to 5 robot morphologies)
> 3.  Faster adaptation (6 hours vs 50+ hours for NavRL)
>
> This suggests that in our setting, decoupling kinematic planning and dynamic adaptation yields better robustness compared to a strong BARN-challenge method (LiCS-KI) and other single-robot baselines, especially in dense obstacle scenarios(No=700: +23% improvement).
>
> ### 2. Necessity of VelFlow & Problem Decomposition
>
> You questioned the complexity of our method and suggested simpler alternatives. During the rebuttal, we systematically evaluated two such "simpler" designs. Both failed to match CE-Nav’s performance, justifying our specific architectural choices:
>
> **Alternative 1: Why not use DWA directly to guide RL? (Skipping Stage 1)**
>
> This fails for two reasons:
> * **Computational BottoleNeck:** DWA is a CPU-intensive search algorithm. Running it at every timestep for 1,024 parallel RL environments is prohibitive, increasing training time from 6 hours to over 4 weeks. VelFlow acts as a GPU-accelerated "neural proxy," enabling massive parallelism.
> * **Resolving "Disastrous Averaging":** DWA outputs a single deterministic command. In ambiguous scenarios (e.g., an obstacle directly ahead), DWA jumps discontinuously between "go left" and "go right." An RL agent guided by such flipping targets tends to average them, leading to a "go straight" collision. VelFlow captures the continuous bimodal distribution, allowing the agent to consistently sample valid modes.
>
> **Alternative 2: Why not fine-tune the VelFlow model directly? (Merging Stages)**
>
> We tested initializing the policy with the pre-trained VelFlow and fine-tuning it end-to-end with RL, rather than using our "Freeze + Refine" approach.
> * **Result:** **Training Collapse**. This setup failed catastrophically, yielding an mSR of only 0.1635 (vs. 0.8575 for CE-Nav).
> * **Reason (Catastrophic Interference):** Navigation requires two conflicting capabilities: Global Geometric Safety and Local Dynamic Feasibility. When fine-tuning end-to-end, the RL gradients (driven by maximizing speed and stability) destroy the delicate geometric priors learned in Stage 1.
>
> **Conclusion:** The novelty of CE-Nav lies in decomposing "complex dynamic navigation" into two solvable sub-tasks:
> 1.  **VelFlow (General Expert):** Handles *Universal Geometric Reasoning*, providing **multimodal** proposals (e.g., "go left OR go right").
> 2.  **RL Refiner (Specific Adapter):** Handles *Embodiment-Specific Dynamics*, providing a **unimodal** correction.
>
> This separation transforms an intractable optimization landscape into a stable one. The Refiner uses the robot's specific dynamics to "collapse" the expert's multimodal probability distribution into a single, executable command.

---

> ### Author Response · Authors · 2025-11-20
> **Response to Reviewer oUen (2/3)**
>
> ### 3. Novelty of "Principled Deviation" & Hybrid Loss
>
>
> We thank the reviewer for highlighting these insightful works. We will explicitly cite and discuss InterMimic [1], DAPG [2], and TWiRL [3] in the revised manuscript. While our hybrid loss formulation
> ($L_{total} = L_{RL} + \lambda L_{BC}$) shares structural similarities with these methods, CE-Nav differs fundamentally in its mechanism for leveraging expert knowledge to handle multi-modality.
>
> **Core Insight:** Conditional Refinement vs. Ambiguity Resolution
>
> **Preserving Multi-modality via Conditioning:**
>
> Navigation is inherently multi-modal (e.g., "pass left" or "pass right"). Our Stage 1 (VelFlow) accurately captures this by learning the conditional probability density p(v∣s).
>
> Crucially, our Stage 2 (RL Refiner) does not attempt to distill this distribution into a single deterministic policy π(s). Instead, the RL policy is conditioned on a specific command sampled from the expert distribution.
>
> This means the RL agent is not forced to "resolve" or "average" conflicting expert modes (which typically leads to collision). Instead, it learns to execute and physically refine whichever specific mode is proposed by the expert (e.g., if the expert samples "left", the RL refines "left"; if "right", it refines "right" under dynamic constraints). This mechanism explicitly preserves the multi-modal nature of the expert's decisions while ensuring dynamic feasibility.
>
> **Contrast with Prior Works:**
>
> InterMimic [1], DAPG [2] and TWiRL [3] typically aim to learn a single deterministic policy π(s) from demonstrations, which destroy the multi-modality. As noted in InterMimic, "ambiguity arises when multiple teachers produce different actions for similar motions." and their RL fine-tuning is explicitly designed to "resolve conflicts"—essentially forcing the policy to converge to a single optimal behavior or averaging out variations.
>
> CE-Nav avoids this "conflict resolution" trap. By structurally decoupling the generation of geometric modes (VelFlow) from dynamics-aware execution (RL Refiner), we solve the "disastrous averaging" problem often encountered in direct imitation or distillation for navigation. The expert provides the choice (distribution), and the RL provides the capability to execute that choice on a specific embodiment.
>
>
> ### 4. Clarification on "Retraining" and Zero-Shot Claims
>
> We agree that "zero-shot" is ideal, but we must distinguish between **Zero-Shot Planning** and **Physical Adaptation**.
>
> * **True Zero-Shot vs. High-Quality Adaptation:** Current zero-shot methods often compromise performance to achieve generality:
>     1.  **Non-hierarchical methods (e.g., X-Nav [4]):** To maintain universality, these methods often sacrifice motion quality on specific platforms. This results in issues like body jitter and unstable gaits (as observed in the X-Nav demo video at 40s-60s: [https://youtu.be/flDyg1Dbna0](https://youtu.be/flDyg1Dbna0)). Our design takes a different stance: we accept a short adaptation phase to recover high-quality behavior on each platform.
>     2.  **Hierarchical Velocity methods:** "Zero-shot" here implies relying on an idealized low-level controller. However, as we demonstrate in **Appendix Table 7**, real-world controllers exhibit drastic differences in tracking error, ranging from **0.28m to 1.56m**. To guarantee safety and robustness in complex, cluttered environments, **adaptation** to these specific dynamics is physically necessary, not just an algorithmic preference.
> * **Not "Retraining from Scratch":** Stage 1 (General Expert) *is* trained once and deployed zero-shot. Stage 2 is strictly for this necessary dynamic adaptation. Adapting to a new robot takes only **6 hours**, compared to **50+ hours** for methods like NavRL or pure RL (Table 1&2). This 8x speedup represents a significant practical advantage.

---

> ### Author Response · Authors · 2025-11-20
> **Response to Reviewer oUen (3/3)**
>
> ### 5. Success Rate & Real-World Complexity
>
> * **Context of SR=0.76:** The 76% success rate cited is from our **most extreme** adversarial setting ($N_o=700$, Table 1). LiCS-KI (the top-performing learning-based algorithm from BARN Challenge) achives the SR=0.58 further proves the difficulty of our evaluation setting. Our mSR is **85.75%**, significantly outperforming NavRL baseline (which drop to 69.25%).
> * **Sim-to-Real Robustness:** We deliberately made the simulation adversarial (denser than typical reality) to stress-test the algorithm. Our real-world success rate is **91.67%** (Table 4), validating that the policy is highly effective in practice.Furthermore, **to directly address the concern that our initial demos were in "simple, open environments,"** we conducted an additional stress test in a **highly constrained physical maze**.
>     * **Experimental Setup:** We constructed a complex obstacle layout featuring **multiple continuous right-angle turns** and **narrow corridors** where the effective width was restricted to be only slightly wider than the robot's body—meaning any slight deviation would result in collision or entrapment.
>     * **Result:** As demonstrated in our new supplementary video, the Unitree Go2 equipped with CE-Nav exhibited excellent performance, navigating these tight confines smoothly. This experiment strongly proves that CE-Nav is not limited to open spaces but is robust enough to handle **highly constrained and cluttered** environments, fully validating its Sim-to-Real generalization capabilities under extreme conditions.
>     * **Video Evidence:** As online playback is not supported, please download the file to view this experiment: [https://anonymous.4open.science/r/1-F649/1.mp4](https://anonymous.4open.science/r/1-F649/1.mp4)
>
> ### 6. Response to Technical Questions
>
> **Q1: Why Normalizing Flow vs. Rectified Flow (RF)?**
> > While Rectified Flow (RF) is a promising generative framework, we selected Conditional Normalizing Flows (CNF/Real-NVP) to prioritize real-time control frequency without sacrificing precision. To quantitatively address your suggestion, we implemented and evaluated two RF variants against our method during rebuttal: standard RF (requiring multi-step ODE solving) and Reflowed RF (distilled for 1-step generation).
>
> | Method | Sampling Steps | Inference Time (ms) | mSR |
> | :--- | :---: | :---: | :---: |
> | **Ours** | 1 | 0.01 | **0.8575** |
> | Rectified Flow | 10 | 0.07 | 0.8300 |
> | Reflowed RF | 1 | 0.007 | 0.8150 |
>
> Latency vs. Precision: Standard RF (10 steps) is approx. 7X slower than our method. While Reflowed RF achieves extreme speed (0.007ms), it suffers a significant drop in performance (0.815 mSR), likely due to distillation errors capturing the complex multimodal velocity distribution.
> Optimal Balance: Our CNF-based approach achieves the highest success rate (0.8575) because of the exact likelihood evaluation, while maintaining negligible inference latency (0.01ms). We will include this ablation study in the revised paper.
>
> **Q2: Why not Waypoints + MPC?**
> > *Velocity vs. Waypoints:* We chose velocity control over Waypoints+MPC because hierarchical waypoint planning often suffers when the low-level controller cannot perfectly track the path. By outputting velocities directly, our Refiner learns to compensate for the robot's specific tracking errors in a closed loop, which is essential for the "Safety" and "Smoothness" objectives.
>
> We hope these clarifications demonstrate that CE-Nav provides a pragmatic, scalable, and robust solution to the Cross-Embodiment Navigation problem. We are committed to incorporating these discussions into the final manuscript.
>
> [1] InterMimic: Towards Universal Whole-Body Control for Physics-Based Human-Object Interactions
>
> [2] Learning Complex Dexterous Manipulation with Deep Reinforcement Learning and Demonstrations
>
> [3] Learning and Adapting Agile Locomotion Skills by Transferring Experience
>
> [4] X-Nav: Learning End-to-End Cross-Embodiment Navigation for Mobile Robots
>
> [5] https://github.com/damanikjosh/the-barn-challenge

---

### Author Response · Authors · 2025-11-27

**Dear Reviewers,**

We succinctly express our sincere gratitude to all reviewers for your detailed and insightful feedback. We are deeply encouraged by the recognition of the **originality of our hierarchical integration**, our **cross-embodiment capability**, the effectiveness of **VelFlow** in resolving multi-modal ambiguity and the successful real-world deployment.

In response to the specific concerns raised by each reviewer, we have provided comprehensive clarifications and conducted extensive new experiments in our individual responses. We have revised the manuscript to incorporate these new updates.

**As the rebuttal phase is ending soon, we welcome any timely discussion if you have remaining questions regarding our work.**

Once again, thank you for your time and efforts in reviewing our paper.

**Best regards,**
**The Authors**

---

### Author Response · Authors · 2025-12-01
**To AC: Summary of Rebuttal**

**Dear Area Chair,**

We sincerely thank the reviewers for their constructive feedback. We are deeply encouraged that **Reviewer Ko6r** recognized the **originality of our hierarchical integration decoupling kinematics and dynamics**, **Reviewer 3iX8** confirmed our **cross-embodiment capabilities**, **Reviewers Ko6r and rjvc** highlighted **VelFlow’s effectiveness** in resolving multi-modal ambiguity, and **Reviewer oUen** acknowledged our **real-world deployment**.

Given the review revert, we summarize the **critical empirical evidence** added during rebuttal:

#### **1. Decisively Surpassing SOTA Baselines (Addressing Reviewer oUen)**

Addressing concerns on baselines, we implemented **LiCS-KI** (the top-performing learning-based algorithm in the BARN Challenge).

* **Result:** CE-Nav significantly outperforms LiCS-KI in dense environments ($N_o=700$), achieving **0.81 SR vs. 0.58 SR** (a **+23% improvement**). On average, we achieve a **+11.3% gain** across all levels.
* **Conclusion:** We provide **substantial gains** over established benchmarks while demonstrating **14.5x faster data generation** and **8x faster adaptation (6h vs 50h for NavRL)**, effectively addressing baseline and efficiency concerns.

#### **2. Validating Architectural Necessity & Novelty (Addressing Reviewer oUen & Ko6r)**
We prove that our **"Guided-RL"** design is necessary:
* **Curriculum Logic (Novel Insight):** Unlike works treating expert data as "Ground Truth," CE-Nav treats it as a **"Physically Idealized Prior."** We decay guidance to encourage **deviating from geometric priors to adapt to specific robot dynamics**.
* **Addressing Covariate Shift:** As illustrated in **Figure 4** (a typical "obstacle encounter" scenario where left/right bypass is geometrically equivalent), the Refiner **does not blindly mimic the expert's symmetrical structure**. Instead, it **independently learns distinct dynamic execution strategies** for each mode to accommodate its own physical characteristics.
* **vs. End-to-End Fine-tuning:** Initializing with VelFlow and fine-tuning end-to-end caused **training collapse (mSR: 0.16)**. Direct fine-tuning collapses the multi-modal distribution to a **single mode**, erasing **multi-modal geometric priors**.
* **vs. Residual RL:** Standard additive Residual RL suffered a **-12.2% performance drop**, confirming additive corrections fail on complex dynamics. **In contrast, our Expert-Conditioned design effectively models these non-linear mappings.**

#### **3. Robustness & Generalization Stress Tests (Addressing Reviewer Ko6r & 3iX8)**
Addressing generalization to diverse morphologies:
* **"Giant Robot" Stress Test:** In a controlled variable test, we modified the Unitree Go2 collision volume to a **1.0m radius** (5x larger). The Refiner successfully adapted (**SR: 0.78**), proving it learns actual physical boundaries.
* **Size Adaptation:** Despite morphological differences (Spot: 1.1m long vs. H1: 0.57m wide), we maintain high success rates.
* **Sim-to-Real in Constrained Spaces:** We demonstrate the Unitree Go2 navigating a **highly constrained maze**, validating robustness.
    * **Video Evidence:** [https://anonymous.4open.science/r/1-F649/1.mp4](https://anonymous.4open.science/r/1-F649/1.mp4)
    * *(Note: As online playback is not supported, please download the file to view this experiment.)*

#### **4. Failure Analysis & Speed-Safety Trade-off (Addressing Reviewer 3iX8)**
We decomposed failure modes to show intelligent adaptation:
* **Agile Robots (Go2):** 0% Timeout, failures due to extreme geometric complexity.
* **Dynamics-Limited Robots (H1):** 9% Timeout. The Refiner learns a **conservative strategy** to compensate for large tracking errors, effectively trading speed for safety.

#### **5. Theoretical Grounding & Real-Time Feasibility (Addressing Reviewer rjvc)**
* **Soundness:** We validated VelFlow is robust to planner bias: **(1)** Mixed-Expert training yielded identical performance; **(2)** Varying DWA radius by $\pm 50\%$ caused negligible drops (**<1%**). We also provided a theoretical **Conformal Mapping** proof for scaling safety.
* **Generative Model Choice:** VelFlow is **7x faster** than Standard Rectified Flow and outperforms both Standard and Reflowed RF variants. Crucially, compressing Diffusion Policy (DP) to 5 steps caused a **-13% performance drop**, validating VelFlow as the optimal edge choice.

**Conclusion**
Rebuttal evidence confirms CE-Nav is a **robust, theoretically grounded, and SOTA-beating framework**. Through our **novel decoupling of kinematic reasoning and dynamic adaptation**, we successfully resolve **"Disastrous Averaging"** via VelFlow and **"Dynamic Mismatch"** via the Refiner. By demonstrating **high efficiency** and **strong generalization** verified in **real-world experiments**, we trust these results effectively address the reviewers' concerns and reinforce the value of our proposed framework.

Sincerely,
The Authors

---

### Meta-Review · Area_Chair_S5HL · 2026-01-10

**Summary:**

This paper proposes CE-Nav, a two-stage framework for cross-embodiment local navigation. A conditional normalizing-flow expert (VelFlow) is trained on large-scale planner data to model multi-modal, geometry-consistent velocity commands, and a lightweight RL refiner then adapts these proposals to the dynamics of each robot. The method is evaluated across multiple morphologies and in real-world deployments.

Here is a summary of the main concerns and how the rebuttal addressed them.

1) Novelty vs. prior demonstration+RL and residual methods.  Reviewers questioned whether the approach goes beyond existing pipelines. The rebuttal provided direct ablations: additive residual RL drops performance by over 10%, and end-to-end fine-tuning of the generative prior collapses. These results support the paper’s key claim that conditioning on a frozen, multi-modal expert is qualitatively different from residual or fine-tuned baselines.

2) Planner bias and covariate shift.  Because VelFlow is trained on DWA data, reviewers worried about bias and the poor standalone performance of the expert. The authors trained mixed-planner experts (DWA+TEB) and obtained essentially identical results, indicating the model learns traversability rather than planner artifacts. Action-distribution analyses further showed how the RL refiner reshapes symmetric geometric modes into dynamically feasible commands, explaining how covariate shift is resolved.

3) Generalization across embodiments.  Concerns about large shape and dynamics differences were addressed with new stress tests, including wide-body humanoids, long quadrupeds, aerial robots, and a 1-meter-radius “giant” robot. In all cases, the refiner learned to override overly optimistic expert proposals and maintained high success rates.

4) Hardware realism and real-time feasibility.  The rebuttal added real-world experiments in narrow, cluttered corridors, frequency- and latency-robustness tests, and embedded-hardware comparisons showing that VelFlow achieves a better speed–performance trade-off than diffusion variants at practical control rates.

Overall, the rebuttal directly addressed the substantive technical concerns and materially strengthened the evidence for the method.

Recommendation: Accept.

**Reviewer Concerns:**

Here is a summary of the main concerns and how the rebuttal addressed them.

1) Novelty vs. prior demonstration+RL and residual methods.  Reviewers questioned whether the approach goes beyond existing pipelines. The rebuttal provided direct ablations: additive residual RL drops performance by over 10%, and end-to-end fine-tuning of the generative prior collapses. These results support the paper’s key claim that conditioning on a frozen, multi-modal expert is qualitatively different from residual or fine-tuned baselines.

2) Planner bias and covariate shift.  Because VelFlow is trained on DWA data, reviewers worried about bias and the poor standalone performance of the expert. The authors trained mixed-planner experts (DWA+TEB) and obtained essentially identical results, indicating the model learns traversability rather than planner artifacts. Action-distribution analyses further showed how the RL refiner reshapes symmetric geometric modes into dynamically feasible commands, explaining how covariate shift is resolved.

3) Generalization across embodiments.  Concerns about large shape and dynamics differences were addressed with new stress tests, including wide-body humanoids, long quadrupeds, aerial robots, and a 1-meter-radius “giant” robot. In all cases, the refiner learned to override overly optimistic expert proposals and maintained high success rates.

4) Hardware realism and real-time feasibility.  The rebuttal added real-world experiments in narrow, cluttered corridors, frequency- and latency-robustness tests, and embedded-hardware comparisons showing that VelFlow achieves a better speed–performance trade-off than diffusion variants at practical control rates.

**Reviewer Scores:**

Reviewer oUen: likely +2
The rebuttal added the missing SOTA baseline (LiCS-KI), showed large gains in dense BARN-style environments, and ruled out the reviewer’s proposed simpler alternatives (direct DWA guidance, end-to-end fine-tuning, additive residual RL). These directly target the reviewer’s main objections about baselines, complexity, and unclear benefit of VelFlow.

Reviewer rjvc: likely +1
The rebuttal addressed planner bias (mixed-planner training), embodiment mismatch (large-robot stress test and conformal scaling argument), and real-time concerns (frequency and latency robustness). These resolve most of the reviewer’s stated weaknesses.

Reviewer Ko6r: likely +1
The authors ran nearly all requested ablations: DWA hyperparameters, diffusion vs flow on embedded hardware, embodiment-scale stress tests, state-space coverage, and cluttered real-world experiments. This substantially strengthens both novelty and empirical support.

Reviewer 3iX8: likely +1
Added analyses of robot size, narrow environments, failure modes, and constrained real-world tests directly answer the reviewer’s concerns while reinforcing their already positive assessment.

---

### Decision · Program_Chairs · 2026-01-26

Accept (Poster)